# DeltaGNN: Graph Neural Network with Information Flow Control

## Abstract

Graph Neural Networks (GNNs) are popular machine learning models designed to process graph-structured data through recursive neighborhood aggregations in the message passing process. When applied to semi-supervised node classification, the message-passing enables GNNs to understand short-range spatial interactions, but also causes them to suffer from over-smoothing and over-squashing. These challenges hinder model expressiveness and prevent the use of deeper models to capture long-range node interactions (LRIs) within the graph. Popular solutions for LRIs detection are either too expensive to process large graphs due to high time complexity or fail to generalize across diverse graph structures. To address these limitations, we propose a mechanism called *information flow control*, which leverages a novel connectivity measure, called *information flow score*, to address over-smoothing and over-squashing with linear computational overhead, supported by theoretical evidence. Finally, to prove the efficacy of our methodology we design DeltaGNN, the first scalable and generalizable approach for long-range and short-range interaction detection. We benchmark our model across 10 real-world datasets, including graphs with varying sizes, topologies, densities, and homophilic ratios, showing superior performance with limited computational complexity.

## 1 Introduction

GNNs are machine learning models designed to process graph-structured data (Scarselli et al., 2008). They have proven effective in various graph-based downstream tasks (Kipf & Welling, 2016; Wu et al., 2019), especially in semi-supervised node representation learning (Hu et al., 2019). GNNs also demonstrate broad applicability across several distinct domains, including chemistry (Gilmer et al., 2020; Zitnik & Leskovec, 2017), and medical fields (Ahmedt-Aristizabal et al., 2021), such as network neuroscience (Bessadok et al., 2015), and medical image segmentation for purposes such as liver tumor and colon pathology classification (Yang et al., 2023).

The strength of GNN models lies in their ability to process graph-structured data and capture short-range spatial node interactions through local neighborhood aggregations during message passing. However, such local aggregation paradigm often struggles to capture dependencies between distant nodes, particularly in certain graph densities and structures. For instance, when processing a strongly clustered graph, a standard GNN model may fail to account for interactions between nodes in distant clusters. These long-range interactions (LRIs) are crucial for node classification tasks, as they help distinguish between different classes and improve classification accuracy. This limitation has been widely studied, with theoretical findings pinpointing over-smoothing (Li et al., 2018) and over-squashing (Alon & Yahav, 2020) as the root causes. These phenomena limit performance in many applications and prevent the use of deep GNNs to effectively capture long-range dependencies.

To address these challenges, recent works have proposed enhanced GNN models by integrating graph transformer-based modules, such as global attention mechanisms (Wu et al., 2021), or local-global attention (Fei et al., 2023). Other works have proposed topological pre-processing techniques, such as curvature-enhanced edge rewiring (Nguyen et al., 2023) and sequential local edge rewiring (Barbero et al., 2023). Although these methods can marginally improve model performance, they fail to provide a generalized and scalable approach that can effectively process large graphs across various graph topologies. Specifically, attention-based approaches are constrained by

their quadratic time complexity and lack of topological awareness, which leads to computational inefficiency. Rewiring algorithms, on the other hand, rely on expensive connectivity measures that are often impractical for large and dense graphs. These methods are also embedding-agnostic and, consequently, fail to directly address over-smoothing in certain graph structures. For these reasons, developing a scalable and general method for learning long-range interactions (LRIs) would significantly expand the applicability of GNNs in semi-supervised node classification tasks. Such a method would be a key contribution to the field.

In this work, we formalize the concept of *graph information flow* and use it to define a novel connectivity measure that analyzes the velocity and acceleration of node embedding updates during message passing, offering insights into graph topology and homophily. We provide both theoretical and empirical evidence to support this claim. Next, we propose a new graph rewiring paradigm, *information flow control*, which mitigates both over-smoothing and over-squashing with minimal additional time and memory complexity. Furthermore, we introduce a novel GNN architecture, *DeltaGNN*, which implements information flow control to capture both long-range and short-range interactions. We benchmark our model across a wide range of real-world datasets, including graphs with varying sizes, topologies, densities, and homophilic ratios. Finally, we compare the results of our approach with popular state-of-the-art methods to evaluate its generalizability and scalability.

**Our contributions.** (1) We introduce a novel connectivity measure, the *information flow score*, which identifies graph bottlenecks and heterophilic node interactions, supported by both theoretical and empirical evidence of its efficacy. (2) We propose an *information flow control* mechanism that leverages this measure to perform sequential edge-filtering with linear computational overhead, which can be flexibly integrated into any GNN architecture. (3) We design a scalable and generalizable framework, *DeltaGNN*, for detecting both short-range and long-range interactions, demonstrating the effectiveness of our theoretical findings.

## 2 PRELIMINARIES

A graph is usually denoted as $\mathcal{G} = (\mathcal{V}, \mathcal{E})$, where $\mathcal{V}$ represents the node set and $\mathcal{E}$ the edge set. The edge set can also be represented as an adjacency matrix, defined as $\boldsymbol{A} \in \{0, 1\}^{|\mathcal{V}| \times |\mathcal{V}|}$, where $\boldsymbol{A}_{ij} \neq 0$ if and only if the edge $(i, j)$ exists. The node feature matrix is defined as $\boldsymbol{X} \in \mathbb{R}^{|\mathcal{V}| \times d_{\mathcal{V}}}$ for some feature dimensions $d_{\mathcal{V}}$. We denote $\boldsymbol{X}_u^t$ as the features of the node $u$ at layer $t$, with the convention $\boldsymbol{X}^0 = \boldsymbol{X}$. Furthermore, each node $u$ is associated with a class $y_u \in \mathcal{C}$, where $\mathcal{C}$ denotes the set of classes. The goal, in node-classification tasks, is to learn $\Phi : \mathcal{V} \to \mathcal{C}$, the unique function mapping each node to a specific class.

### 2.1 GRAPH NEURAL NETWORKS

Each layer of the GNN applies a transformation function and message-passing aggregation function to each feature vector $\boldsymbol{X}_u$ and its neighborhood $\mathcal{N}(u)$. The general formulation of this operation can be expressed as follows:

$$\boldsymbol{X}_u^{t+1} = \phi \left( \bigoplus_{v \in \tilde{\mathcal{N}}(u)} \psi(\boldsymbol{X}_v^t) \right) \quad \text{for } 0 \leq t \leq T \tag{1}$$

where $\phi$ and $\psi$ are differentiable functions, $\oplus$ is an aggregation function (typically sum, mean, or max), $\tilde{\mathcal{N}}(u) = \mathcal{N}(u) \cup u$ the extended neighbourhood of $u$, and $T$ is the number of layers of the model.

### 2.2 OVER-SMOOTHING AND HOMOPHILY

Over-smoothing (Rusch et al., 2023) can be formalized as follows.

**Definition 2.1** (Over-smoothing). *Over-smoothing refers to the phenomenon where the representations of nodes become indistinguishable as the number of layers $T$ increases, weakening the expressiveness of deep GNNs and limiting their applicability. In node classification tasks, over-smoothing*

*can hinder the ability of GNNs to distinguish between different classes, as their feature vectors converge to a fixed value with increasing layers:*

$$\sum_{(u,v)\in\mathcal{E}:\Phi(u)\neq\Phi(v)} ||\boldsymbol{X}_u^T - \boldsymbol{X}_v^T|| \to 0 \; as \; T \to \infty$$

Over-smoothing is accelerated by the presence of edges connecting distinct node classes (Chen et al., 2020), these are called heterophilic edges. The tendency of a graph to lack heterophilic connections is termed homophily. This property can be quantified computing the graph homophilic ratio $\mathcal{H} \in [0, 1]$, which is the average of the homophilic ratios of its nodes.

**Definition 2.2** (Homophilic Ratio of a Node). *The homophilic ratio $\mathcal{H}_u \in [0, 1]$ of a node $u$ is the proportion of neighbours $u' \in \mathcal{N}(u)$ such that $u$ and $u'$ belong to the same class. The homophilic ratio is defined as:*

$$\mathcal{H}_u = \frac{|\{v \in \mathcal{N}(u) \mid \Phi(v) = \Phi(u)\}|}{|\mathcal{N}(u)|}$$

Over-smoothing can be mitigated by increasing graph homophily, for instance, by removing heterophilic edges.

### 2.3 OVER-SQUASHING AND CONNECTIVITY

Over-squashing (Alon & Yahav, 2020), on the other hand, is the inhibition of the message-passing capabilities of the graph caused by graph bottlenecks.

**Definition 2.3** (Over-squashing). *Over-squashing refers to the exponential growth of a node's receptive field, leading to the collapse of substantial information into a fixed-sized feature vector due to graph bottlenecks (see Figure 1).*

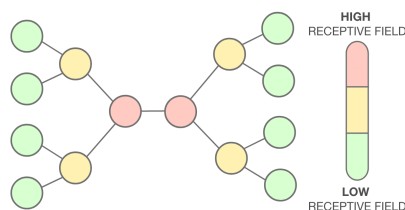

Figure 1: Illustration of a graph with a bottleneck.

To alleviate over-squashing, it is necessary to improve the connectivity of the graph. This can be achieved by removing or relaxing graph bottlenecks that hinder connectivity (Di Giovanni et al., 2023). Typically, these bottlenecks can be identified by computing a connectivity measure, such as the Ollivier-Ricci curvature (Ollivier, 2009; Sia et al., 2019) or a centrality measure. The term *connectivity measure* refers to any topological or geometrical quantity that captures how easily different pairs of nodes can communicate through message passing.

## 3 RELATED WORKS

**Transformer-based self-attention.** A common approach to overcome over-smoothing and over-squashing and capture long-range node interactions is integrating a transformer-based components into the GNN (Yun et al., 2019; Dwivedi & Bresson, 2020; Fei et al., 2023; Wu et al., 2021). Transformers can aggregate information globally without being limited by the local neighborhood aggregation paradigm, making them a very effective solution. However, they fail to propose a scalable solution which can process large-scale graphs, which is arguably the scenario where long-range interaction detection is of the utmost importance. Graph self-attention has a time complexity of $O(|\mathcal{V}|^2)$, where $|\mathcal{V}|$ refers to the number of nodes in the graph, this complexity is unsuitable for large graphs. Moreover, classical transformers are inefficient at processing LRIs in graphs because they are inherently *topology-agnostic* and consequently process all $|\mathcal{V}|^2$ possible interactions.

**Topological graph rewiring.** Many rewiring algorithms exploit graph curvature (Nguyen et al., 2023) or other connectivity measures (Barbero et al., 2023; Black et al., 2023; Arnaiz-Rodríguez et al., 2022; Karhadkar et al., 2022) to identify parts of the graph suffering from over-squashing and

over-smoothing, and alleviate these issues by adding or removing edges. These approaches have two main limitations: first, they rely on expensive connectivity measures (e.g., Ollivier-Ricci curvature, betweenness centrality); and second, they are *embedding-agnostic*, as they primarily base rewiring decisions on the graph's connectivity. However, over-smoothing is not a topological phenomenon, and as observed by recent works (Yan et al., 2022), its effects can be mitigated by acting on the graph's heterophily. Therefore, improving connectivity alone is insufficient to prevent over-smoothing (see Appendix A).

# 4 INFORMATION FLOW

## 4.1 GRAPH INFORMATION FLOW

We define *Graph Information Flow* (GIF) as the exchange of information between nodes during the message-passing process. This can be quantified as the rate at which node embeddings are aggregated across each layer of the GNN. To observe how GIF changes over time within our model, we introduce two sequences that are useful for quantifying this variation. Let $M$ be a smooth manifold equipped with a distance function $d : M \times M \to \mathbb{R}$ that defines the geometry of the space. Additionally, assume the features embeddings $\mathbf{X}$ of the input graph lie on $M$.

**Definition 4.1** (First Delta Embeddings). *Let $\mathbf{M}_u^t = \psi(\mathbf{X}_u^{t-1})$ be the transformed feature vector of node $u$ at layer $t$. We define the first delta embeddings at layer $t$ as the distance between the aggregated and transformed feature vectors.*

$$\Delta_u^t = d\big( \bigoplus_{v \in \mathcal{N}(u)} \mathbf{M}_v^t, \mathbf{M}_u^t \big) \quad for \ 1 \le t \le T$$

*Where $T \in \mathbb{N}$ defines the number of layers of the architecture. This quantity can be interpreted as the velocity at which the node embeddings are aggregated at layer $t$.*

**Definition 4.2** (Second Delta Embeddings). *Similarly, let $\Delta_u^t$ be the first delta embedding of a node $u \in \mathcal{V}$ at time $t \in [1, T]$, where $T \in \mathbb{N}$ defines the number of layers of the architecture. Then, we define the second delta embeddings as the first differences of the first delta embeddings over time.*

$$(\Delta^2)_u^t = d(\Delta_u^t, \Delta_u^{t-1}) \quad for \ 2 \le t \le T \quad and \quad (\Delta^2)_u^1 = 0$$

*This can be interpreted as the rate of change in the rate at which node embeddings are aggregated within the model, analogous to acceleration in physical systems.*

We now provide theoretical evidence showing how the sequences $\Delta_u^t$ and $(\Delta^2)_u^t$ can offer insights into the graph's homophily and topology, respectively. Detailed proofs of the following lemmas can be found in Appendices B and C.

**Lemma 1.** *Let $\Delta_u^t$ be the first delta embeddings of a node $u$ and $\overline{\Delta_u}$ be the average over time of the sequence. Assume $\Delta_u^t$ converges to zero, $M$ is compact and that there exists a unique function $\phi : M \to C$ which correctly assign all possible feature vectors to their associated labels. Then, for any homophilic ratio $\mathcal{H} \in [0, 1]$, there exists a positive lower-bound $\rho \in (0, +\infty)$ such that any node $u \in \mathcal{V}$ with feature vector $\mathbf{X}_u \in M$ and $\overline{\Delta_u} > \rho$ will have $\mathcal{H}_u < \mathcal{H}$.*

**Lemma 2.** *Let $c : \mathcal{V} \to \mathbb{R}$ be a node connectivity measure, and let $\mathbb{V}_t[\Delta_u^2]$ denote the variance over time of the second delta embeddings of a node $u$. Assume that there exists an upperbound $\mu$ such that for any node $u \in \mathcal{V}$, $c(u) < \mu$ if and only if the node $u$ is adjacent to an edge bottleneck. Then, any node $u \in \mathcal{V}$ for which $c(u) < \mu$ will exhibit a low value of the variance $\mathbb{V}_t[\Delta_u^2]$.*

## 4.2 INFORMATION FLOW SCORE

While over-squashing is a topological phenomenon caused by graph bottlenecks, over-smoothing is worsened due to the presence of heterophilic node interactions. Motivated by our ultimate goal of mitigating these phenomena, and building on the results of Lemmas 1 and 2, we define a node connectivity measure, called the *information flow score (IFS)*, which is minimized for nodes near bottlenecks with a low homophilic ratio (see Equation 2). Consequently, nodes with low values

of this measure are likely to correspond to regions where over-smoothing and over-squashing may occur.

$$\mathbf{S}_u = \frac{m * \mathbb{V}_t[\Delta_u^2] + 1}{l * \overline{\Delta_u} + 1} \tag{2}$$

Where ($l$ and $m$) are two multipliers that adjust the weight of the mean and variance in the score, respectively. A detailed analysis justifying our definition of the IFS, along with empirical evidence supporting its properties, can be found in Appendix D.

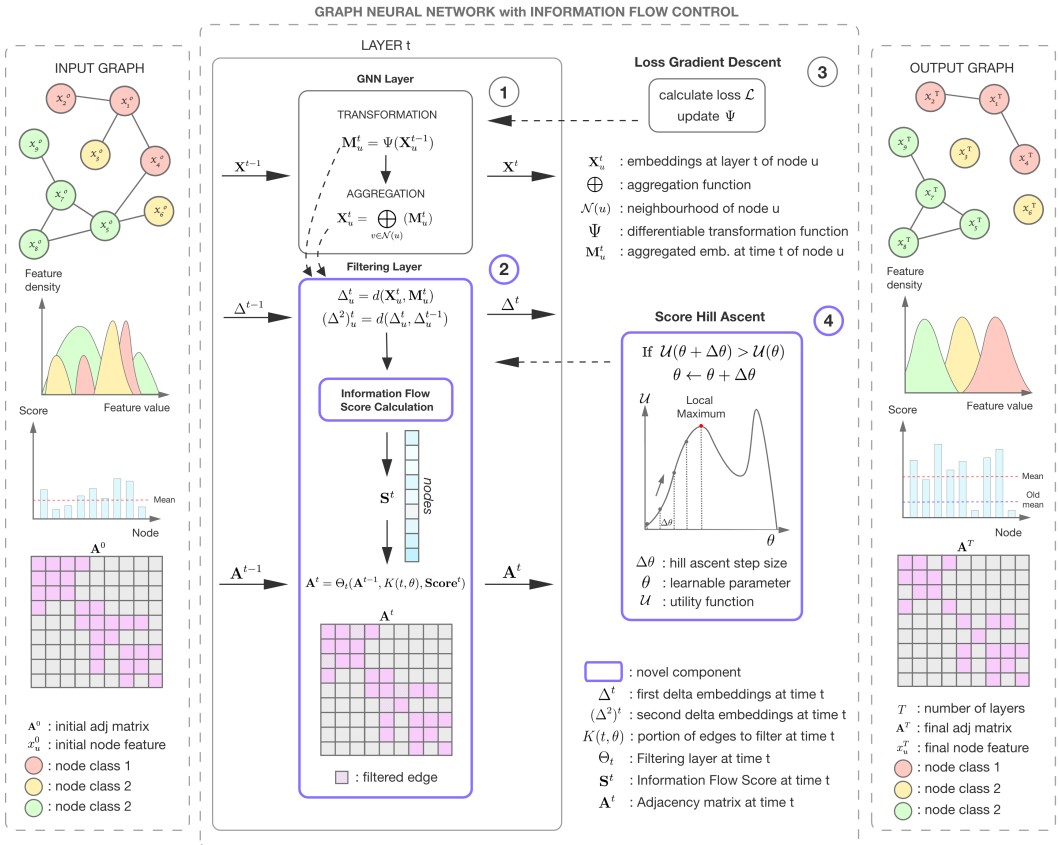

Figure 2: *Overview of a GNN Model with information flow control*. The figure illustrates how the information flow control mechanism integrates with a standard GNN to filter the graph and increase the mean node score. Simultaneously, the GNN learns to disentangle the node features, ensuring that the output embeddings can be easily classified using a readout layer. The novel components are highlighted in violet. Notably, the mean node score of the processed graph is significantly higher due to the effective edge filtering performed within the GNN layers.

### 4.3 INFORMATION FLOW CONTROL

Now, we present how to integrate the IFS into a generic GNN to prevent over-smoothing and over-squashing. While common connectivity measures are typically calculated before transforming the graph, our score is computed during the graph transformation learning. Therefore, standard approaches such as graph-rewiring algorithms are not well-suited for handling our measure. To address this limitation, we designed a novel method called *information flow control (IFC)* (see Figure 2). We first introduce some notations and then define the novel components of the IFC. We define a topological edge-filtering operation as a function $\Theta(\mathcal{G}, K(\theta), c)$ returning a filtered graph $\mathcal{G}'$, where $K(\theta) \in [0, 1]$ is a $\theta$-parameterized function defining the percentage of edges which has to

be removed and $c \in \mathbb{R}^{|\mathcal{V}|}$ are the nodes connectivity values used to choose which edges has to be removed. The IFC is composed of a sequence of $T$ flow-aware edge-filtering layers $\{\Theta_t\}_t$ that iteratively filter the graph $\mathcal{G}$:

$$\mathcal{G} = \mathcal{G}_0 \xrightarrow{\Theta_1} \mathcal{G}_1 \xrightarrow{\Theta_2} \mathcal{G}_2 \dots \xrightarrow{\Theta_T} \mathcal{G}_T$$

Each filtering operation is defined as $\Theta_t(\mathcal{G}_{t-1}, K(t, \theta), \mathbf{S}^t)$, where $\mathbf{S}^t \in \mathbb{R}^{|\mathcal{V}|}$ is the IFS of the graph nodes calculated on the deltas up to layer $t$. These filtering layers are interwoven with standard GNN layers $\{\Omega_t\}_t$, which are responsible to aggregate and transform the node representations.

$$(\mathbf{A}, \mathbf{X}) = (\mathbf{A}^0, \mathbf{X}^0) \xrightarrow{\Omega_1} (\mathbf{A}^0, \mathbf{X}^1) \xrightarrow{\Theta_1} (\mathbf{A}^1, \mathbf{X}^1) \xrightarrow{\Omega_2} (\mathbf{A}^1, \mathbf{X}^2) \dots \xrightarrow{\Theta_T} (\mathbf{A}^T, \mathbf{X}^T)$$

Where $\mathbf{A}^t$ denotes the adjacency matrix of $\mathcal{G}$ at layer $t$. The IFC also includes a component that maximizes the mean node score by optimizing the parameter $\theta$ through hill ascent over a utility function $\mathcal{U}$. We set an initial value of zero for $\theta$ and perform a local hill ascent to find a local maximum. This approach is advantageous because the IFC is encouraged to preserve sparsity, increasing the number of edges removed only if it leads to an increase in the utility function $\mathcal{U}$ (e.g., the mean of the scores at the last layer). Follows a pseudo-code implementing the forward-pass of a GNN with IFC module (additional technical details are included in Appendix E).

---

**Algorithm 1** GNN with information flow control: forward-pass

---

1: **Input:** Node features $\mathbf{X}^0$, adjacency matrix $\mathbf{A}^0$, Previous utility value $\mathcal{U}_{old}$
2: **Output:** Processed features $\mathbf{X}^T$, updated adjacency matrix $\mathbf{A}^T$
3: Initialize variable $\Delta^0 = 0$, score array $\mathbf{S}$, $\theta = 0$
4: **for** each layer $t$ **do**
5:     Compute transformed features $\mathbf{M}^t$                           ▷ equation 1
6:     Compute $\mathbf{X}^t$ by aggregating $\mathbf{M}^t$                   ▷ equation 1
7:     Compute $\Delta^t$                                      ▷ equation 4.1
8:     **if** not first layer **then**
9:         Compute $(\Delta^2)^t$                           ▷ equation 4.2
10:     **else**
11:         Set $(\Delta^2)^t = 0$
12:     **end if**
13:     Update $\overline{\Delta^t}$ applying Welford's Method on $\overline{\Delta^{t-1}}$.     ▷ equation 5
14:     Update $\mathbb{V}[(\Delta^2)^t]$ applying Welford's Method on $\mathbb{V}[(\Delta^2)^{t-1}]$.     ▷ equation 6
15:     Compute scores $\mathbf{S}^t$                             ▷ equation D
16:     Update $\mathbf{A}^t$ removing $K(t, \theta)$ edges with low IFS from $\mathbf{A}^{t-1}$
17: **end for**
18: Compute utility value $\mathcal{U}_{new}$ based on final scores $\mathbf{S}^T$ (e.g. mean node score)
19: **if** $\mathcal{U}_{new} > \mathcal{U}_{old}$ **then**
20:     Update parameter $\theta$ with step size $\Delta\theta$ *(hill ascent step)*
21: **end if**
22: **return** processed features $\mathbf{X}^T$, updated adjacency matrix $\mathbf{A}^T$, new utility value $\mathcal{U}_{new}$

---

## 5 METHODOLOGY: DELTAGNN

In this section, we illustrate DeltaGNN and how it leverages the IFC to capture both short-range and long-range node interactions. First, DeltaGNN processes the node embeddings and graph adjacency matrix through a standard homophilic GNN with IFC. During the sequential filtering, the model partitions the graph into homophilic clusters by removing bottlenecks and heterophilic edges. Simultaneously, the homophilic aggregation learns short-range dependencies. However, this process leads to the loss of long-range interactions, which we recover through an heterophilic graph condensation. The long-range dependencies are then learned via a GNN heterophilic aggregation. Finally, a readout layer processes the results from both GNN aggregations. An overview of the model pipeline is illustrated in Figure 3.

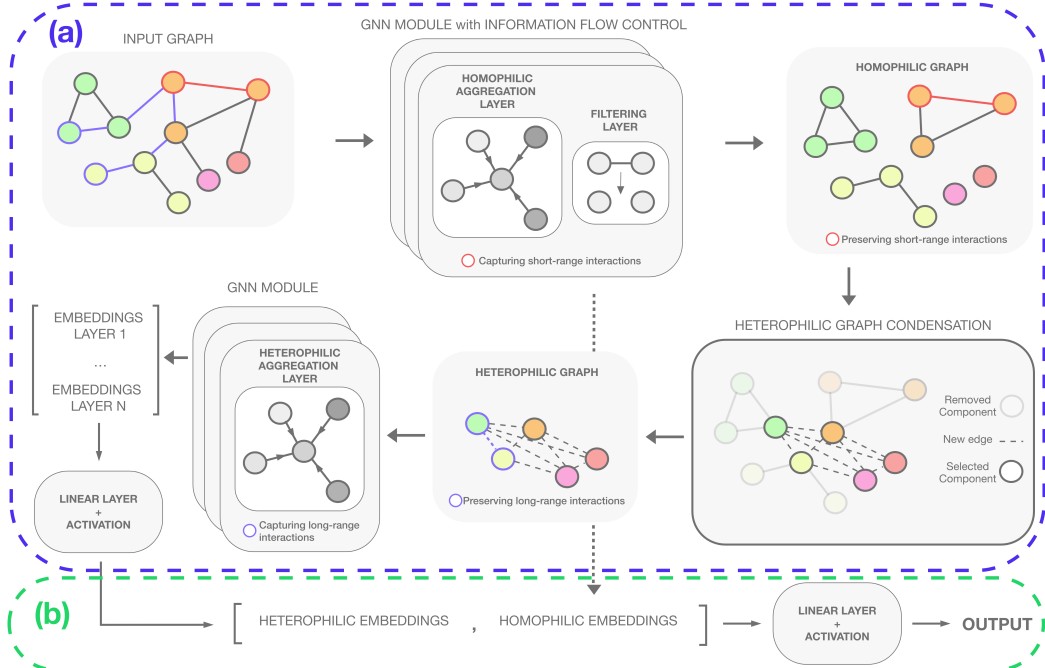

Figure 3: The DeltaGNN pipeline consists of: (a) a *sequential transformation* stage that processes both homophilic and heterophilic edge interactions, performing homophily-based interaction-decoupling and dual aggregation to learn short-term and long-term dependencies; and (b) a *prediction* stage that concatenates the results from the first stage.

**Sequential transformation stage.** The *sequential transformation* stage includes a *homophilic aggregation with IFC*, a *heterophilic graph condensation*, and a *heterophilic aggregation*. This stage is designed to discriminate between homophilic and heterophilic node interactions, producing a strongly homophilic graph and a strongly heterophilic graph, which are processed by independent GNN modules. This concept of homophily-based interaction-decoupling is crucial to prevent over-smoothing by avoiding using a standard GNN aggregation on heterophilic edges. During the homophilic aggregation (see Equation 1) the model capture short-range spatial interactions, while the IFC removes most heterophilic edges within the graph, breaking it into homophilic connected components (see Figure 3). We set $\mathcal{U} = \overline{\mathbf{S}^T}$ and the step size $\Delta\theta = \eta$, where $\eta$ denotes the learning rate of the transformation function of the GNN module. At each GNN layer $t$, we remove the $K(t, \theta)$ edges with the lowest scores, which are calculated using the Euclidean distance $d(\mathbf{x}, \mathbf{y}) = \|\mathbf{x} - \mathbf{y}\|$. This process reduces over-smoothing by increasing the homophily of the graph, and over-squashing, by removing the graph bottlenecks. As a result, the output graph of this first step, which we define $\mathcal{G}_{ho}$, will be highly homophilic and preserve most short-term node interactions. On the other hand, most long-range interaction will be lost during the edge filtering. Next, we proceed with the *heterophilic graph condensation*, which aims at extracting the most relevant heterophilic interactions within the homophilic clusters to re-introduce long-range dependencies. To achieve this, we select the nodes with the highest scores, which are likely to have the most reliable representations, and construct a distinct fully-connected graph with them. The resulting graph, defined as $\mathcal{G}_{he}$, will be highly heterophilic since nodes with different labels are likely to be neighbours due to the clustering during the score-based edge filtering. This graph will preserve the majority of the LRIs while resulting considerably smaller than the original graph. Next, we apply the *heterophilic aggregation* on $\mathcal{G}_{he}$ which is a simple variation of the standard update rule defined in equation 3.

$$\boldsymbol{X}_u^{t+1} = \phi\left(\bigoplus_{v \in \mathcal{N}(u)} \psi(\boldsymbol{X}_v^t), \psi^s(\boldsymbol{X}_u^t)\right), \quad \boldsymbol{X}_u^1 = \phi\left(\psi^s(\boldsymbol{X}_u^0)\right), \quad \text{for } 0 \le t \le L-1 \tag{3}$$

where $\psi^s$ is a differentiable function used to process node self-connections. Additionally, we do not aggregate the node features in the first layer so that the model can process the original features. Finally, we build a row vector of all layers' node embedding outputs $\mathbf{X}_{he}^i$ and feed it into a linear layer, similarly to what was done in Xu et al. (2018b) (see Equation 4). This ensures that the model learns to distinguish between different node classes at many levels of smoothness.

**Prediction stage.** In this stage, we concatenate and process both modules' outputs, $\boldsymbol{X}_{ho}^{out}$ and $\boldsymbol{X}_{he}^{out}$, through a final linear layer to perform the prediction (see Equation 4).

$$\boldsymbol{X}_{he}^{out} = \phi_{he} \left( \boldsymbol{W}_{he}^{out} \begin{bmatrix} \boldsymbol{X}_{he}^1 \\ \boldsymbol{X}_{he}^2 \\ \vdots \\ \boldsymbol{X}_{he}^T \end{bmatrix} \right) \quad , \quad \boldsymbol{X}^{out} = \phi_{out} \left( \boldsymbol{W}^{out} \begin{bmatrix} \boldsymbol{X}_{ho}^{out} \\ \boldsymbol{X}_{he}^{out} \end{bmatrix} \right) \tag{4}$$

## 6 Experiments

In this work, we evaluate DeltaGNN on 10 distinct datasets with varying homophilic ratios, densities, sizes, and topologies. For all datasets, the hyper-parameter fine-tuning was done using a grid-search on the validation set. We compare our DeltaGNN (implemented using GCN aggregations), with state-of-the-art GNN architectures such as GCN (Kipf & Welling, 2016), GIN (Xu et al., 2018a), GAT (Velickovic et al., 2017), rewiring algorithms (Topping et al., 2021), heterophily-based methods (Zhu et al., 2020; Pei et al., 2020), and graph-transformers (Shi et al., 2020; Chen et al., 2022). Additional details on dataset details, training settings, and models can be found in the Appendix F.

- **Cora, CiteSeer, and PubMed datasets.** We evaluate our framework on three scientific publications-citations datasets (McCallum et al., 2000; Yang et al., 2016) with high homophilic ratios ($\geq 0.5$). These datasets include one graph each and the task is node-level multi-class classification across several fields of research.

- **Cornell, Texas, and Wisconsin datasets.** We also use three webpage network datasets (Pei et al., 2020) with low homophilic ratios ($\leq 0.5$). These datasets include one graph each and the task is node-level multi-class classification among several webpage categories.

- **MedMNIST Organ-C and Organ-S datasets.** The MedMNIST Organ-C and Organ-S datasets (Yang et al., 2023) include abdominal CT scan images ($28 \times 28$ pixels) of liver tumors in coronal and sagittal views, respectively. The task for both is node-level multi-class classification among 11 types of liver tumors. Each image-based dataset is converted into a graph, with each node representing an image, and each node embedding being the vectorization of the 28 by 28 pixel intensity, resulting in vectors of length 784. The graph edges are derived from the cosine similarity of the node embeddings, similar to what as done in Adnel & Rekik (2023). To reduce complexity, we sparsify these graphs using a sparsity threshold and convert them into unweighted graphs. Additionally, we define two degrees of density for each dataset, with up to $\sim 2.8$ million edges.

### 6.1 Generalizability Benchmark

In this section, we assess the efficacy of our models on six datasets with varying homophilic ratios. As most GNNs rely on graph homophily, this experiment helps us gauge our model dependence on this assumption and its generalizability across different domains. As shown in Table 1, DeltaGNN outperforms all state-of-the-art methods across four out of six datasets, with an average accuracy increase of **1.23%**. When comparing the performance of different connectivity measures, we notice that both topology-based and geometry-based connectivity measures fail to offer a generalizable solution across different homophilic scenarios. On the other hand, DeltaGNN is the only approach that consistently yields the best results, consolidating our claim that the IFS is a one-for-all connectivity measure. Furthermore, while expensive measures such as betweenness centrality (BC), closeness centrality (CC), and Ollivier-Ricci curvature (RC) encountered out-of-time (OOT) errors on the PubMed dataset, our IFS, along with degree centrality (DC), eigenvector centrality (EC), and Forman-Ricci curvature (FC), proved scalable enough to process all six datasets. When analyzing the results obtained from datasets with low homophilic rates, such as Texas and Wisconsin, we

observe that the performance gap between DeltaGNN and rewiring approaches increases, demonstrating the effectiveness of IFC in improving the homophilic rate of a graph through iterative edge filtering. A more rigorous experiment on this observation is presented in the Appendix G, where we compare the rate of change of the graph homophilic ratio during the edge filtering using different connectivity measures.

Table 1: Accuracy $\pm$ std over 5 runs on six datasets with varying homophily.

| Methods | CORA $\mathcal{H} = 0.81$ | Citeseer $\mathcal{H} = 0.74$ | PubMed $\mathcal{H} = 0.80$ | Cornell $\mathcal{H} = 0.30$ | Texas $\mathcal{H} = 0.09$ | Wisconsin $\mathcal{H} = 0.19$ |
|---|---|---|---|---|---|---|
| GCN (Kipf & Welling, 2016) | 85.12±0.42 | 76.10±0.32 | **88.26±0.48** | 37.84±2.70 | 70.81±2.96 | 55.69±2.63 |
| GCN + *rewiring(DC)* | 85.32±0.49 | **76.36±0.38** | 88.24±0.40 | 41.08±1.21 | 70.81±3.52 | 56.47±1.64 |
| GCN + *rewiring(EC)* | 84.92±0.52 | 75.80±0.46 | 88.24±0.25 | 50.54±0.00 | 69.19±4.10 | 52.16±5.65 |
| GCN + *rewiring(BC)* | 84.82±0.71 | 75.42±0.62 | OOT | 37.30±4.44 | 65.94±5.27 | 54.90±2.78 |
| GCN + *rewiring(CC)* | 85.06±0.26 | 75.44±0.51 | OOT | 37.30±6.45 | 69.73±2.96 | 54.90±2.78 |
| GCN + *rewiring(FC)* | 85.12±0.45 | 75.48±0.60 | 87.94±0.33 | 50.54±0.00 | 67.57±5.06 | 52.55±3.22 |
| GCN + *rewiring(RC)* | 84.90±0.62 | 75.94±0.43 | OOT | 31.35±11.40 | 71.34±2.42 | **56.86±4.80** |
| GCN + *rewiring(IFS)* | **85.36±0.21** | 75.92±0.22 | 87.66±0.88 | **50.54±0.00** | **71.35±1.48** | 54.51±3.51 |
| GIN (Xu et al., 2018a) | 83.84±0.70 | 72.72±0.73 | 87.80±0.42 | **58.38±2.42** | 56.21±5.86 | 52.55±0.88 |
| GIN + *rewiring(DC)* | 83.76±0.43 | 72.84±0.59 | 87.94±0.64 | 48.11±7.50 | 60.54±4.52 | 53.33±2.15 |
| GIN + *rewiring(EC)* | **84.02±0.85** | **73.56±0.28** | **88.20±0.77** | 50.27±6.22 | 60.54±3.63 | 52.94±3.67 |
| GIN + *rewiring(BC)* | 83.66±0.79 | 73.02±0.58 | OOT | 55.67±4.10 | 58.38±4.10 | **55.29±3.51** |
| GIN + *rewiring(CC)* | 83.60±0.70 | 72.90±0.99 | OOT | 55.13±4.91 | 65.40±4.83 | 53.72±2.23 |
| GIN + *rewiring(FC)* | 83.16±0.93 | 73.50±0.84 | 87.30±0.37 | 56.22±5.54 | 58.92±4.83 | 49.80±2.97 |
| GIN + *rewiring(RC)* | 83.22±0.72 | 73.32±1.28 | OOT | 57.84±3.63 | **61.62±4.44** | 52.94±2.77 |
| GIN + *rewiring(IFS)* | 82.94±1.00 | 73.38±1.30 | 87.90±0.96 | 54.95±5.86 | 57.84±1.48 | 53.33±2.91 |
| GAT (Velickovic et al., 2017) | 85.42±0.95 | 78.08±0.26 | 85.48±0.58 | 38.38±4.83 | 64.32±2.26 | 51.37±1.64 |
| GAT + *rewiring(DC)* | 85.26±0.67 | 77.82±0.33 | 85.36±0.52 | 37.30±4.44 | 62.16±5.06 | 50.59±3.51 |
| GAT + *rewiring(EC)* | 85.30±0.87 | 77.54±0.31 | 85.34±0.57 | **38.92±4.52** | 60.00±5.20 | 50.59±1.64 |
| GAT + *rewiring(BC)* | 85.46±1.21 | 78.26±0.50 | OOT | 38.92±4.52 | **64.86±4.27** | 52.16±4.51 |
| GAT + *rewiring(CC)* | 85.50±0.72 | 78.10±0.59 | OOT | 37.30±4.83 | 63.78±3.08 | **52.16±2.23** |
| GAT + *rewiring(FC)* | 86.00±0.87 | 77.86±0.77 | **85.60±0.27** | 37.30±4.01 | 58.92±10.71 | 50.20±4.29 |
| GAT + *rewiring(RC)* | 85.28±0.57 | **78.64±0.36** | OOT | 35.13±3.31 | 62.70±6.73 | 51.76±1.07 |
| GAT + *rewiring(IFS)* | **85.80±0.40** | 78.06±0.78 | 85.02±0.58 | 35.13±4.27 | 57.84±11.40 | 47.84±4.07 |
| MPL (LeCun et al., 2015) | 70.32±2.68 | 68.64±1.98 | 86.46±0.35 | 71.62±5.57 | 77.83±5.24 | 82.15±6.93 |
| SDRF (Topping et al., 2021) | 86.40±2.10 | 72.58±0.20 | OOT | 57.54±0.34 | 70.35±0.60 | 61.55±0.86 |
| H2GCN (Zhu et al., 2020) | 83.48±2.29 | 75.16±1.48 | 88.86±0.45 | 75.40±4.09 | 79.73±3.25 | 77.57±4.11 |
| GEOM-GCN (Pei et al., 2020) | 84.10±1.12 | 76.28±2.06 | 88.13±0.67 | 54.05±3.87 | 67.57±5.35 | 68.63±4.92 |
| UniMP (Shi et al., 2020) | 84.18±1.39 | 75.00±1.59 | 88.56±0.32 | 66.48±12.5 | 73.51±8.44 | 79.60±5.41 |
| NAGphormer (Chen et al., 2022) | 85.77±1.35 | 73.69±1.48 | 87.87±0.33 | 56.22±8.08 | 63.51±6.53 | 62.55±6.22 |
| DeltaGNN - *control* | 84.56±0.57 | 79.40±0.77 | 89.64±0.73 | 75.13±1.21 | 67.57±2.70 | 74.12±1.64 |
| DeltaGNN - *control + DC* | 84.60±1.05 | **79.90±0.79** | 89.70±0.10 | **75.67±1.91** | 72.43±1.21 | 76.47±1.39 |
| DeltaGNN - *control + EC* | 84.14±0.63 | 79.36±0.59 | 89.68±0.47 | 72.97±3.31 | 73.51±1.21 | 74.90±3.22 |
| DeltaGNN - *control + BC* | 84.36±0.57 | 78.98±0.86 | OOT | 72.97±1.91 | 70.81±3.52 | 74.51±2.77 |
| DeltaGNN - *control + CC* | 84.54±0.93 | 79.46±0.75 | OOT | 74.05±1.48 | 70.81±2.26 | 75.29±2.97 |
| DeltaGNN - *control + FC* | 84.94±0.75 | 79.36±0.65 | **89.98±0.24** | 73.51±1.21 | 71.89±1.48 | 73.33±1.07 |
| DeltaGNN - *control + RC* | 84.96±0.50 | 79.34±0.59 | OOT | 74.05±1.48 | 72.43±1.91 | 76.08±0.88 |
| DeltaGNN *constant* | 86.38±0.18 | 79.15±0.43 | 89.73±0.31 | 70.27±4.10 | **74.05±3.08** | 79.21±1.75 |
| DeltaGNN *linear* | **87.29±0.52** | 79.42±0.78 | 89.60±0.45 | 75.27±3.31 | 72.97±3.82 | **80.00±0.88** |

**Notes**: Results better than their counterparts have a more intense shade of green. OOT indicates an out-of-time error (compute time $\geq$ 30 mins).

## 6.2 SCALABILITY BENCHMARK

DeltaGNN exhibits the lowest epoch time in three out of five datasets, in some cases being up to three times faster than its ablations (see Table 7). The introduction of IFC in DeltaGNN resulted in a reduction of the average epoch time by **30.61%** when compared to the worst-performing model (see Table 2). However, DeltaGNN shows higher-than-average epoch time on the CiteSeer dataset due to the large size of feature vectors. Additionally, thanks to IFC, DeltaGNN is the only LRI architecture with no preprocessing overhead. In terms of memory consumption, DeltaGNN uses approximately twice the memory of standard GCN models. This is due to its implementation, which employs GCN aggregation layers along with dual homophilic and heterophilic aggregations, inherently requiring more parameters. Nevertheless, GAT showed the highest memory footprint, being the only model that failed to process both dense datasets. Graph-transformers and heterophily-based methods proved to be at least as computationally expensive as GAT, failing to process the largest

graphs. For this reason, in this benchmark we only compare rewiring approaches. In Appendix D.1, we provide an in-depth analysis of the time and memory complexity of our methodology.

Table 2: Computational resource comparison. The best connectivity measure with respect to epoch time is used. The rate of change is calculated relative to the worst-performing variation.

| Methods | CORA | | CiteSeer | |
|---|---|---|---|---|
| | Δ GPU Memory | Δ Epoch Time | Δ GPU Memory | Δ Epoch Time |
| GCN + *filtering* | **-69.99%** | -31.62% | **-63.49%** | **-40.90%** |
| GIN + *filtering* | -25.58% | -30.84% | -30.72% | -40.26% |
| GAT + *filtering* | -55.10% | 0% | -54.11% | 0% |
| DeltaGNN - control | 0% | -19.63% | 0% | -33.48% |
| DeltaGNN | 0% | **-42.06%** | 0% | -14.64% |

| Methods | PubMed | | Organ-S | |
|---|---|---|---|---|
| | Δ GPU Memory | Δ Epoch Time | Δ GPU Memory | Δ Epoch Time |
| GCN + *filtering* | **-65.68%** | -6.69% | **-70.65%** | -17.27% |
| GIN + *filtering* | -52.53% | -6.62% | -60.28% | -17.21% |
| GAT + *filtering* | 0% | -3.49% | 0% | -6.93% |
| DeltaGNN - control | -16.48% | 0% | -30.78% | 0% |
| DeltaGNN | -16.48% | **-56.16%** | -29.47% | **-24.10%** |

| Methods | Organ-S *dense* | |
|---|---|---|
| | Δ GPU Memory | Δ Epoch Time |
| GCN + *filtering* | **-57.63%** | **-41.50%** |
| GIN + *filtering* | -44.19% | -41.45% |
| GAT + *filtering* | OOM | OOM |
| DeltaGNN - control | -3.20% | 0% |
| DeltaGNN | 0% | -16.11% |

**Notes**: Results that outperform their counterparts are shaded with a more intense green. OOM indicates an out-of-memory error.

# 7    CONCLUSION

In this work, we introduced the concepts of *information flow score (IFS)* and *information flow control (IFC)*, novel approaches for mitigating the effects of over-smoothing and over-squashing during message passing in semi-supervised node classification tasks. To demonstrate the effectiveness of our methodology, we developed DeltaGNN, the first GNN architecture to incorporate IFC for detecting both short-range and long-range node interactions. We provided rigorous theoretical evidence and extensive experimentation to support our claims. Our empirical results show that our methodology outperforms popular state-of-the-art methods. As a future direction, we plan to explore alternative implementations of DeltaGNN with different aggregation paradigms and non-GNN components and examine the applicability of our approach to graph-level and edge-level learning tasks.

# 8    REPRODUCIBILITY STATEMENT

Our implementation of the proposed methods and the scripts to reproduce the experiments are publicly available at https://anonymous.4open.science/r/DeltaGNN-DA28/. The experimental settings, including dataset details, preprocessing techniques, evaluation models, hyperparameter values, fine-tuning techniques, and hardware and software configurations, are provided in Appendix F. Additional details on the implementation can be found in Appendix E.

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

## A  GRAPH-REWIRING ALGORITHMS ON OVER-SMOOTHING

In this section, we demonstrate why common graph-rewiring algorithms fail to address over-smoothing in certain graph structures. Typically, these algorithms utilize a connectivity measure to detect dense areas within the graph, sparsifying them to slow down over-smoothing, while relaxing bottlenecks by adding new edges to reduce over-squashing. We independently illustrate these two rewiring strategies on a small graph in Figure 4.

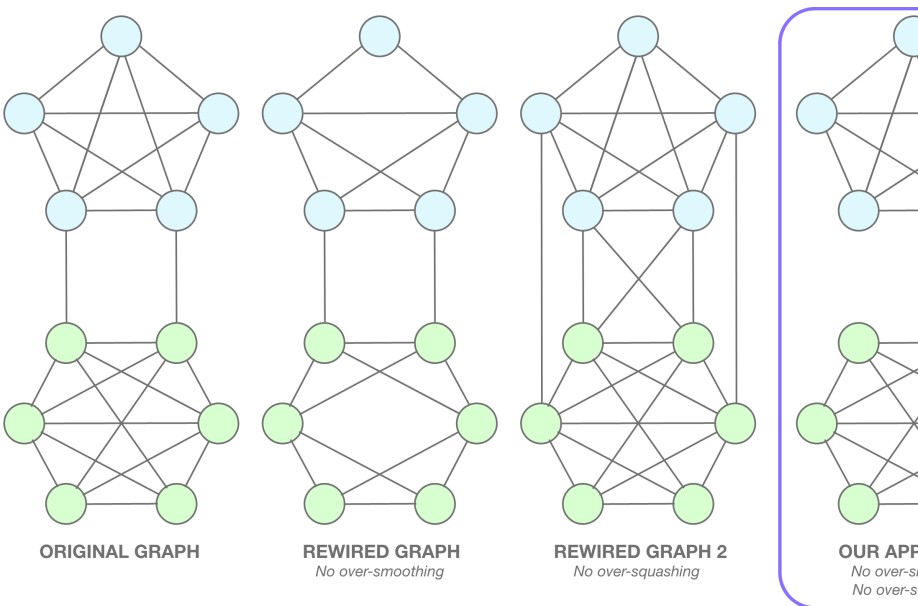

Figure 4: Comparison of graph-rewiring techniques for alleviating over-smoothing and over-squashing. Distinct node colors represent different node classes.

In general, reducing the number of edges in a graph slows the convergence of node representations during message-passing. Popular rewiring algorithms exploit this fact and aim to sparsify dense areas of the graph, which are the first to experience smoothing during neighborhood aggregation. This approach merely slows down the aggregation process in the GNN, postponing the onset of over-smoothing while also decelerating the node representation learning. To directly mitigate over-smoothing, feature aggregation between nodes of different classes must be prevented by removing heterophilic edges. However, relaxing bottlenecks by adding new edges can significantly decrease the homophilic ratio of the graph and exacerbate over-smoothing (see Figure 4). This explains why many rewiring algorithms either fail to address or even worsen the performance of the underlying models.

While common rewiring algorithms rely on connectivity measures (e.g., centrality or curvature-based measures) that consider only the graph topology, ignoring node embeddings and thus graph homophily, our method leverages the rate of change in node embeddings to detect heterophilic edges and successfully prevent over-smoothing.

## B  PROOF OF LEMMA 1

To prove Lemma 1, we first introduce two Theorems.

**Theorem B.1.** *Hopf–Rinow Theorem (Hopf & Rinow, 1931):*
*Let $(M,d)$ be a connected Riemannian manifold with a distance function $d$ defining the geometry of the space. The following conditions are equivalent:*

    *1. $(M,d)$ is geodesically complete, i.e., every geodesic can be extended indefinitely.*

2. *(M,d) is a complete metric space with respect to the distance function induced by the Riemannian metric d.*

3. *Any closed and bounded subset of $M$ is compact.*

4. *For any two points p, q in $M$, there exists a minimizing geodesic between p and q.*

**Theorem B.2.** *Extreme Value Theorem (Rusnock & Kerr-Lawson, 2005):*
*Let $f : [a, b] \to \mathbb{R}$ be a continuous function defined on a closed interval $[a, b]$. Then, $f$ attains both a maximum and a minimum value on $[a, b]$; that is, there exist points $x_{min}, x_{max} \in [a, b]$ such that:*

$$f(x_{min}) \leq f(x) \leq f(x_{max}) \quad \text{for all } x \in [a, b].$$

**Lemma 1.** *Let $\Delta_u^t$ be the first delta embeddings of a node $u$ and $\overline{\Delta_u}$ be the average over time of the sequence. Assume $\Delta_u^t$ converges to zero, $M$ is compact and that there exists a unique function $\phi : M \to C$ which correctly assign all possible feature vectors to their associated labels. Then, for any homophilic ratio $\mathcal{H} \in [0, 1]$, there exists a positive lower-bound $\rho \in (0, +\infty)$ such that any node $u \in \mathcal{V}$ with feature vector $\mathbf{X}_u \in M$ and $\overline{\Delta_u} > \rho$ will have $\mathcal{H}_u < \mathcal{H}$.*

*Proof.* To prove the lemma, we first show that there exists $\rho$, which is a valid positive lower-bound for every $\Delta_u^t$ for some node $u$. Then, we deduce that $\rho$ is also a valid lower bound for $\overline{\Delta_u}$.

For any node $u \in \mathcal{V}$ within the graph, with feature vector $\mathbf{X}_u^t \in M$, homophilic ratio $\mathcal{H}_u$, and neighbourhood $\mathcal{N}(u) = \{n_1, n_2, ..., n_k\}$ of degree $k$, we define $S$ as the family of all feature assignments $s : \mathcal{N}(u) \to M$ mapping each neighboring node of $u$ to a feature vector in $M$ such that the following constraint is respected:

$$\mathcal{H}_u = \frac{|\{m \in s(v) : v \in \mathcal{N}(u)) \mid \Phi(m) = \Phi(\mathbf{X}_u^t)\}|}{k}$$

Here, the constraint ensures that the feature assignment respects the given homophily ratio $\mathcal{H}_u$. Thanks to the existence of $\Phi$ we know that the cardinality of $S$ is at least one, since we can always define $S \ni \tilde{s}(n_i) = \mathbf{X}_{n_i}^t \ \forall i \in [1, k]$ where $\mathbf{X}_{n_i}^t$ are columns of the feature matrix $\mathbf{X}^t$ associated to the neighbouring nodes $n_i$. In this case, the constraint will reduce to the definition of homophilic ratio of $u$. Now, we proceed defining $\Delta_u^t$ with respect to a feature assignment $s \in S$ as $\tilde{\Delta}_u^t(s) := d(\bigoplus_{v \in \mathcal{N}(u)} s(v), \mathbf{M}_u^t)$ for which it holds that $\tilde{\Delta}_u^t(\tilde{s}) = \Delta_u^t$. Since our manifold $M$ is compact, the *Hopf–Rinow Theorem* B.1 ensures that the distance metric $d$ is continuous, which also implies that our $\tilde{\Delta}_u^t(s)$ is continuously defined for any feature assignment. By the *Extreme Value Theorem* B.2 and the fact that set $S$ is non-trivial, $\tilde{\Delta}_u^t(s)$ must attain its maximum real value for some feature assignment $s \in S$. As a result, for any node $u$ with a certain homophilic ratio $\mathcal{H}_u$ we can define the real positive constant $U(\mathcal{H}_u)_u := \sup_{s \in S} \tilde{\Delta}_u^t(s)$ as the maximum value that the first delta embeddings at time $t$ can take for any possible feature assignment.

Now, for any homophilic ratio $\mathcal{H} \in [0, 1]$ we can choose a $\rho > \sup_{h \in [\mathcal{H}, 1]} U(h_u)_u$ such that any node $u$ with $\Delta_u^t > \rho$ will have $\mathcal{H}_u < \mathcal{H}$. This last implication can be verified assuming, for the sake of contradiction, that the node $u$ may have $\Delta_u^t > \rho$ and $\mathcal{H}_u \geq \mathcal{H}$, and observing that this leads to the following contradiction:

$$\rho < \Delta_u^t = \tilde{\Delta}_u^t(\tilde{s}) \leq \sup_{s \in S} \tilde{\Delta}_u^t(s) = U(\mathcal{H}_u)_u \leq \sup_{h \in [\mathcal{H}, 1]} U(h_u)_u < \rho$$

Since this relation holds true for any $t$, we can deduce:

$$\Delta_u^t > \rho \to \sum_{i \in [1, T]} \Delta_u^i > T\rho \to \overline{\Delta_u} > \rho$$

where the penultimate inequality holds if and only if the series is convergent, completing the proof. $\square$

In practice, this generalization over the mean helps reduce noise that may affect the first layers. The value of $\rho$ depends on the maximum distance between the subspaces of $M$ associated with the node classes. In some domains, lower thresholds can be obtained, allowing for better distinction between nodes with different homophilic ratios. Finally, since the node classes are hidden from the model, we cannot directly compute the threshold $\rho$. However, it is sufficient to know that, since $\rho$ exists, the nodes with the highest values of $\overline{\Delta}$ will likely have the lowest homophilic rates.

## C  PROOF OF LEMMA 2

**Lemma 2.** *Let $c : \mathcal{V} \to \mathbb{R}$ be a node connectivity measure, and let $\mathbb{V}_t[\Delta_u^2]$ denote the variance over time of the second delta embeddings of a node $u$. Assume that there exists an upperbound $\mu$ such that for any node $u \in \mathcal{V}$, $c(u) < \mu$ if and only if the node $u$ is adjacent to an edge bottleneck. Then, any node $u \in \mathcal{V}$ for which $c(u) < \mu$ will exhibit a low value of the variance $\mathbb{V}_t[\Delta_u^2]$.*

*Proof.* We provide an informal proof of the lemma. Recently, Nguyen et al. (2023) demonstrated that nodes adjacent to bottlenecked edges are less affected by over-smoothing, while nodes located in dense areas of the graph experience faster convergence of their vector embeddings. To prove the lemma, we extend these results by observing that $\mathbb{V}_t[\Delta_u^2]$ can be used to classify a node $u$ into one of these two categories.

This observation follows from the definition of over-squashing. Building on the results of Nguyen et al. (2023), we can formalize the correlation between over-squashing and connectivity as follows: for a pair of nodes $u$ and $v$ with feature vectors $\mathbf{X}_u^t$ and $\mathbf{X}_v^t$, respectively, where $c_u < \mu < c_v$ (with $c_u$ and $c_v$ representing their respective connectivity), and assuming both nodes are equidistant from their neighborhood (implying the same amount of information is aggregated at time $t = 1$), we have $\Delta_v^t \in o(\Delta_u^t)$ (in the little-o notation). In other words, node $v$ experiences faster convergence compared to node $u$, as nodes near bottlenecks tend to have constrained communication paths in the graph and are consequently less likely to experience significant fluctuations in their embedding values. Now, we can distinguish the two cases: $v$ will converge faster, leading to high values of $(\Delta_v^2)^t$ for small values of $t$ and low values later in time. On the other hand, for node $u$, the values of $(\Delta_u^2)^t$ will vary more slowly over time due to its slower convergence. This intuitive observation implies that $\mathbb{V}_t[\Delta_v^2]$ must necessarily be greater than $\mathbb{V}_t[\Delta_u^2]$.

This concludes the proof, as we have shown that a node with a low connectivity measure (or, equivalently, a bottlenecked node) exhibits a relatively low variance of the second delta embeddings compared to other nodes in the graph.

$\square$

We would like to discuss one more point: What happens when we compare nodes belonging to very different neighborhoods? When our assumption of the same distance *w.r.t.* the neighborhoods is not respected, considerable additional noise is introduced into the process, and the convergence of the sequence $\Delta_v^u$ will also depend on other factors, such as node homophily or feature separability. In these cases, it is evident that our approach will fail to be a deterministic solution for detecting graph bottlenecks, and the accuracy will depend on the amount of noise introduced by these external factors.

## D  INFORMATION FLOW SCORE: DETAILED ANALYSIS

We now explain the rationale behind the definition of the *information flow score*:

$$\mathbf{S}_u = \frac{m * \mathbb{V}_t[\Delta_u^2] + 1}{l * \overline{\Delta_u} + 1}$$

We remind that our primary goal is to enhance the message-passing mechanism in our model to better capture long-range interactions by alleviating both over-smoothing and over-squashing. To achieve this, we need a connectivity measure capable of detecting bottlenecks and heterophilic edges, which should be removed to mitigate these phenomena. From Lemma 1, we know that a

high value of $\overline{\Delta}$ is likely associated with a low homophilic ratio, and from Lemma 2, a low $\mathbb{V}_t[\Delta^2]$ indicates proximity to an edge bottleneck. Consequently, the score is defined as a fraction involving these two terms. Since, in some applications, we may prioritize detecting bottlenecks over heterophilic edges (or vice versa), we introduce two multipliers, $l$ and $m$, to adjust the weights of the mean and variance in the final score. In our case, we aim to detect both with equal priority, so we set $m = 1$ and $l = 1$. To ensure the score remains well-defined, regardless of the delta values, and to prevent it from exploding when the deltas are close to zero, we add 1 to both the numerator and denominator. An additional advantage of adding 1 is that isolated nodes will receive a score of one, which is relatively low. As a result, training the model to rewire the graph while maximizing the overall score will encourage the model to remove edges while still preserving sparsity. To reduce the impact of noise affecting the initial delta values, both the variance and mean are computed using an exponential moving average, which assigns greater weight to more recent data points.

### D.1 COMPLEXITY OF INFORMATION FLOW SCORE

The major advantage of using the IFS lies in its synergy with GNNs. Since all GNNs process the input graph through message-passing, it is possible to calculate the score with almost no additional computational cost, as the score for any node has constant time complexity. The additional overhead for computing the score is $O(|\mathcal{V}|d_{\mathcal{V}}T)$, depending on the distance function used. Typically, we have $T \ll d_{\mathcal{V}} \ll |\mathcal{V}|$, which results in an average time complexity of $O(|\mathcal{V}|)$. To the best of our knowledge, this offers the lowest time complexity among all major connectivity measures proposed in the literature (see Table 3).

The memory complexity of the IFS is also linear with respect to the number of nodes, $O(|\mathcal{V}|)$, since both the variance and mean can be computed iteratively using Welford's method (Efanov et al., 2021), without the need to store all previous delta values.

| | **Connectivity Measure** | **Time Complexity** |
|---|---|---|
| Topological Measures | Degree Centrality (DC) | $O(|\mathcal{E}|)$ |
| | Eigenvector Centrality (EC) | $O(|\mathcal{V}| + |\mathcal{E}|)$ |
| | Betweenness Centrality (BC) | $O(|\mathcal{V}||\mathcal{E}|)$ |
| | Closeness Centrality (CC) | $O(|\mathcal{V}||\mathcal{E}|)$ |
| Geometrical Measures | Ollivier-Ricci Curvature (OC) | $O(|\mathcal{V}||\mathcal{E}|)$ |
| | Forman-Ricci Curvature (FC) | $O(|\mathcal{E}|)$ |
| Embeddings-Based Measures | Information Flow Score (IFS) | $O(|\mathcal{V}|)$ |

Table 3: Time complexity of common connectivity measures (Wandelt et al., 2020; Sia et al., 2019) and our novel *information flow score*.

### D.2 EDGE-FILTERING USING INFORMATION FLOW SCORE

We now illustrate how our edge-filtering based on the information flow score can improve the homophily and connectivity of the graph, resulting in an increase in the mean node score. We experiment with a small graph containing fourteen nodes belonging to three distinct classes, including bottlenecked and heterophilic edges. As shown in Figure 5, these edges can be easily detected using the IFS values. We then remove the edges adjacent to the nodes with the lowest scores. The filtered graph, as illustrated in Figure 6, demonstrates higher homophily, fewer bottlenecks, and consequently, a higher mean node score.

## E INFORMATION FLOW CONTROL: TECHNICAL DETAILS

Now, we provide additional technical details regarding the implementation of a generic GNN model with *information flow control*, introducing Welford's method for efficiently calculating the mean and variance of a sequence of variables.

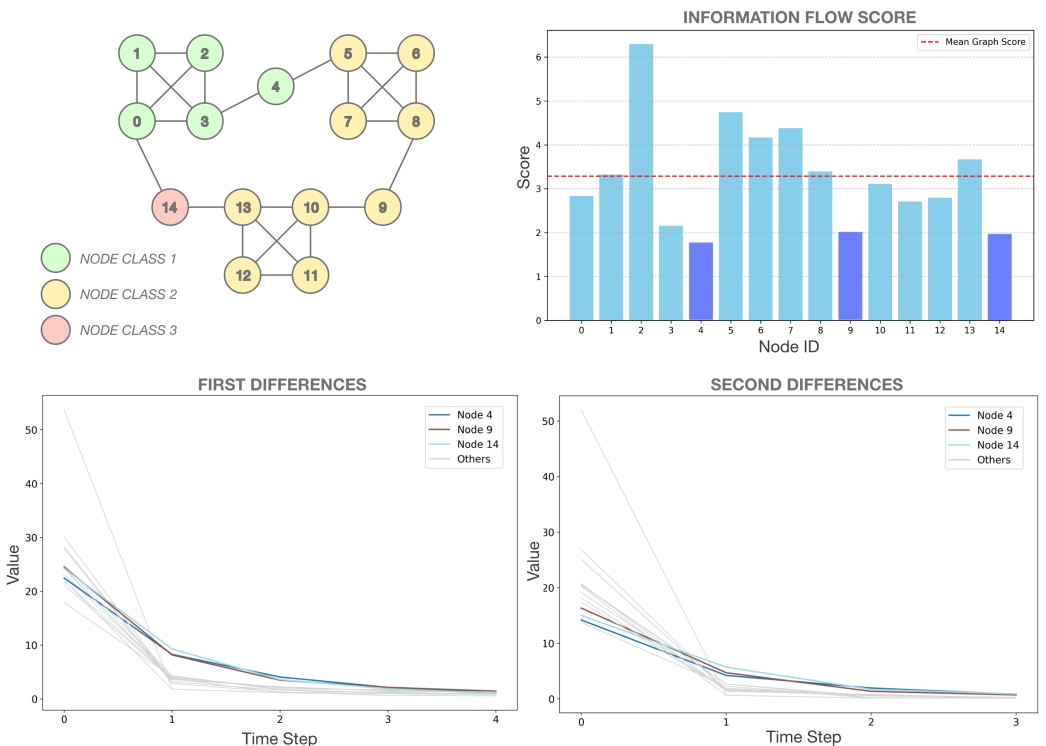

Figure 5: *Illustrations of first delta embeddings, second delta embeddings, and information flow score on a small graph with bottlenecks and heterophilic edges.* We set $m = 1$ and $l = 1$ to detect heterophilic bottlenecks, using the Euclidean distance as the distance metric $d$. The samples used to generate the graph are medical images from the MedMNIST Organ-C dataset. As observed from the plots, nodes 4, 9, and 14 can be easily distinguished as they have very low scores.

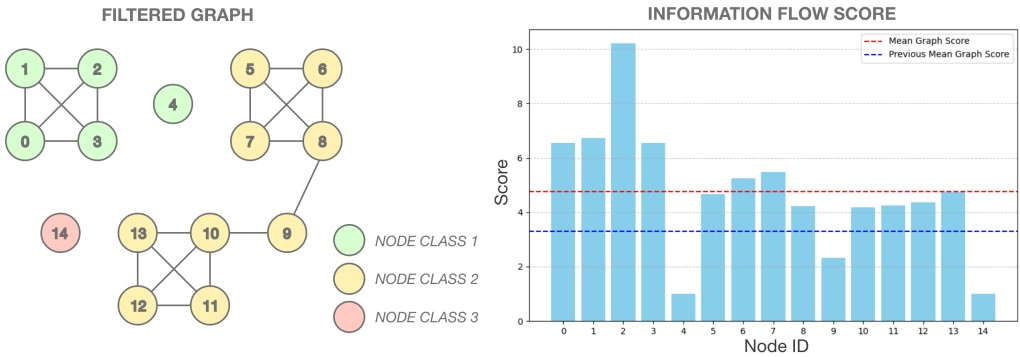

Figure 6: *Filtered graph with updated information flow score.* After removing bottleneck edges and heterophilic node interactions, the mean node score increased significantly, demonstrating the effectiveness of the edge filtering.

### E.1 Welford's Method for Calculating Average and Variance

Given a sequence of numbers $x_1, x_2, \ldots, x_n$, Welford's method calculates the mean $\mu_n$ and variance $\sigma_n^2$. The mean after $n$ elements can be updated incrementally by:

$$\mu_n = \mu_{n-1} + \frac{x_n - \mu_{n-1}}{n} \tag{5}$$

where $\mu_n$ is the mean after the first $(n)$ elements. The variance after $n$ elements can be updated using the following formula:

$$\sigma_n^2 = \sigma_{n-1}^2 + \frac{(x_n - \mu_{n-1}) \times (x_n - \mu_n) - \sigma_{n-1}^2}{n} \tag{6}$$

where $\sigma_n^2$ is the variance after the first $n$ elements.

# F    EXPERIMENTAL DETAILS: DATASETS, MODELS, AND ADDITIONAL RESULTS

In this work, we evaluated DeltaGNN on 10 distinct datasets. All dataset details are summarized in Table 4. This section describes the models used in this benchmark, as well as the hardware and software configurations employed to run the experiments.

## F.1    EVALUATION MODELS

For GCN (Kipf & Welling, 2016), GIN (Xu et al., 2018a), and GAT (Velickovic et al., 2017), we include seven variations that incorporate an edge-rewiring module, which filters edges based on a specific connectivity measure. Specifically, it removes a fixed number of edges from dense areas of the graph to alleviate over-smoothing and removes bottlenecks to prevent over-squashing. We experiment with the following measures: degree centrality (DC), eigenvector centrality (EC), betweenness centrality (BC), closeness centrality (CC), Forman-Ricci curvature (FC), Ollivier-Ricci curvature (RC), and the information flow score (IFS). We also propose two variations of DeltaGNN implementing different functions $K(t, \theta)$; a *constant* function $K = \theta$, and a *linear* function:

$$K = \left( \frac{\theta}{T - 1} \right) \cdot (t - 1)$$

where $T$ denotes the number of layers in the architecture. This last equation describes a line passing through the points $(t = 1, K = 0)$ and $(t = T, K = \theta)$. Using this function, we ensure that fewer edges are removed in the initial layers and increasingly more are removed in the later ones since $K$ is increasing. This is a desirable property of $K$, as we expect the quality of the IFS to improve with an increasing number of aggregations. Additionally, we include an ablation of DeltaGNN without the IFC mechanism, where edge-filtering is done before homophilic aggregation rather than in parallel (resulting in higher time complexity). We also include seven versions of this model, each implementing different connectivity measures.

The hyperparameters used for each model are detailed in Table 5 for reproducibility. To ensure a fair comparison, the number of layers and hidden channels are kept constant across models, with reductions made only in cases of out-of-memory errors. This ensures a balanced comparison of the computational resources used by each method. Fine-tuning has been conducted using a grid-search methodology on the validation set for the following parameters and values: learning rate (0.0001, 0.0005, 0.001, 0.005), edges to remove (values are dataset-specific), maximum number of communities (5, 10, 20, 50, 100, 500), number of layers (3 to 6), and hidden dimensions (100, 256, 512, 1024, 2048).

## F.2    EXPERIMENT CONFIGURATIONS

This section describes the hardware and software configurations used to run the experiments. Our experiments were conducted on a consumer-grade workstation with the following specifications: Intel Core i7-10700 2.90GHz CPU, dual-channel 16GB DDR4 memory clocked at 3200 MHz, and an Nvidia GeForce GTX 1050 Ti GPU with 4GB GDDR5 video memory. The system ran on a Linux-based operating system (Ubuntu 22.04.4 LTS) with NVIDIA driver version 535.183.01 and

Table 4: Dataset details of all datasets.

| | # of Graphs | # of Nodes | # of Edges | # of Features | # of Labels | Task Level | Task Type | Training Type | Training Set Size | Validation Set Size | Test Set Size |
|---|---|---|---|---|---|---|---|---|---|---|---|
| **Planetoid** (Yang et al., 2016) | | | | | | | | | | | |
| Cora $\mathcal{H} = 0.81$ | 1 | 2708 | 5278 | 1433 | 7 | Node | Multi-class | Inductive | 1208 | 500 | 1000 |
| CiteSeer $\mathcal{H} = 0.74$ | 1 | 3327 | 4552 | 3703 | 6 | Node | Multi-class | Inductive | 18217 | 500 | 1000 |
| PubMed $\mathcal{H} = 0.80$ | 1 | 19717 | 44324 | 500 | 3 | Node | Multi-class | Inductive | 1827 | 500 | 1000 |
| **WebKB** (Pei et al., 2020) | | | | | | | | | | | |
| Cornell $\mathcal{H} = 0.30$ | 1 | 183 | 295 | 1703 | 5 | Node | Multi-class | Inductive | 87 | 59 | 37 |
| Texas $\mathcal{H} = 0.09$ | 1 | 183 | 309 | 1703 | 5 | Node | Multi-class | Inductive | 87 | 59 | 37 |
| Wisconsin $\mathcal{H} = 0.19$ | 1 | 251 | 499 | 1703 | 5 | Node | Multi-class | Inductive | 120 | 80 | 51 |
| **MedMNIST** (Yang et al., 2023) | | | | | | | | | | | |
| Organ-S | 1 | 25221 | 1276046 | 784 | 11 | Node | Multi-class | Inductive | 13940 | 2452 | 8829 |
| Organ-S (*dense*) | 1 | 25221 | 2494750 | 784 | 11 | Node | Multi-class | Inductive | 13940 | 2452 | 8829 |
| Organ-C | 1 | 23660 | 1241622 | 784 | 11 | Node | Multi-class | Inductive | 13000 | 2392 | 8268 |
| Organ-C (*dense*) | 1 | 23660 | 2809204 | 784 | 11 | Node | Multi-class | Inductive | 13000 | 2392 | 8268 |

Table 5: Hyperparameter configurations of all datasets.

| | # of Layers | Hidden Channels (Others) | Hidden Channels (GAT) | Hidden Channels (DeltaGNN) | Dropouts | # Removed Edges (Edge-Filter) | # Max Communities | Learning Rate | Optimizer |
|---|---|---|---|---|---|---|---|---|---|
| **Planetoid** (Yang et al., 2016) | | | | | | | | | |
| Cora $\mathcal{H} = 0.81$ | 3 | 2048 | 2048 | 2048 | 0.5 | 40/30 | 20 | 0.0005 | ADAM |
| CiteSeer $\mathcal{H} = 0.74$ | 3 | 2048 | 2048 | 2048 | 0.5 | 40/30 | 20 | 0.0005 | ADAM |
| PubMed $\mathcal{H} = 0.80$ | 3 | 1024 | 256 | 1024 | 0.5 | 400/200 | 10 | 0.0005 | ADAM |
| **WebKB** (Pei et al., 2020) | | | | | | | | | |
| Cornell $\mathcal{H} = 0.30$ | 3 | 2048 | 2048 | 2048 | 0.5 | 10/5 | 20 | 0.0005 | ADAM |
| Texas $\mathcal{H} = 0.09$ | 3 | 2048 | 2048 | 2048 | 0.5 | 10/5 | 20 | 0.0005 | ADAM |
| Wisconsin $\mathcal{H} = 0.19$ | 3 | 2048 | 2048 | 2048 | 0.5 | 10/5 | 20 | 0.0005 | ADAM |
| **MedMNIST** (Yang et al., 2023) | | | | | | | | | |
| Organ-S | 3 | 1024 | 256 | 1024 | 0.5 | 1000/5000 | 500 | 0.0005 | ADAM |
| Organ-S (*dense*) | 3 | 1024 | 256 | 1024 | 0.5 | 3000/15000 | 500 | 0.0005 | ADAM |
| Organ-C | 3 | 2048 | 2048 | 2048 | 0.5 | 1000/5000 | 500 | 0.0005 | ADAM |
| Organ-C (*dense*) | 3 | 2048 | 2048 | 2048 | 0.5 | 3000/15000 | 500 | 0.0005 | ADAM |

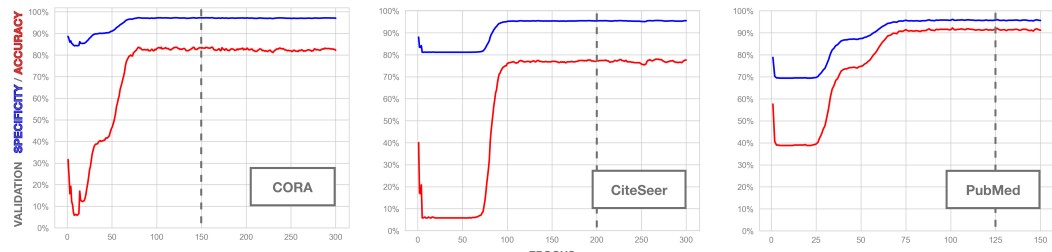

Figure 7: Validation accuracy (red) and specificity (blue) convergence during training epochs for DeltaGNN *linear* on three datasets with varying homophilic ratios. The dashed line indicates the observed convergence point.

CUDA toolkit version 12.2. Our implementation uses Python 3.10.12, PyTorch 2.2.0 (Paszke et al., 2017) (with CUDA 11.8), and Torch Geometric 2.5.1 (Fey & Lenssen, 2019). The results presented are based on 95% confidence intervals over 5 runs, and all datasets were trained inductively. The training pipeline includes an early stopping mechanism with a patience counter to prevent overfitting.

### F.3 Additional Experiments: size-varying and density-varying graphs

When processing large and dense graphs, we observe a significant performance improvement with DeltaGNN compared to GAT (Velickovic et al., 2017), GIN (Xu et al., 2018a), and GCN (Kipf & Welling, 2016). DeltaGNN consistently delivered the best results across all four datasets. Additionally, while GAT encountered out-of-memory errors, and most connectivity measures resulted in out-of-time errors when processing the two *dense* datasets, our proposed models implementing the IFS measure successfully completed all benchmarks and achieved the best average performance. Overall, DeltaGNN performed well on the MedMNIST (Yang et al., 2023) datasets, with an average accuracy increase of **+0.92%**. Graph-transformers and heterophily-based methods proved to be at least as computationally expensive as GAT, also failing to process the largest graphs. For this reason, and due to hardware limitations, we only compare rewiring approaches and standard GNN baselines in this benchmark (see Table 6).

## G Connectivity Measure Comparison

In this section, we directly compare the efficacy of different connectivity measures during topological edge-filtering and heterophilic graph condensation. Figure 8 and Table 8 illustrate how the density distribution of the node homophilic rate changes throughout these two stages using various connectivity measures on the CORA dataset. Following the example of recent works (Mao et al., 2024), we define the homophilic ratio to account for both feature similarity and label similarity to more accurately depict the performance disparity between different methods. Let $e$ be an edge connecting the nodes $u$ and $v$ with feature vectors $\mathbf{X}_u$ and $\mathbf{X}_v$, respectively. Let $\Phi$ be a function mapping each feature vector to its associated label, and $\delta$ a function returning 1 if the two inputs coincide and 0 otherwise. Then the homophilic rate of $e$ is defined as:

$$\mathcal{H}_e = 0.5 \cdot \frac{\mathbf{X}_u \cdot \mathbf{X}_v}{\|\mathbf{X}_u\|\|\mathbf{X}_v\|} + 0.5 \cdot \delta(\Phi(\mathbf{X}_u), \Phi(\mathbf{X}_v))$$

where the left term denotes the cosine similarity of the two feature representations.

During topological edge-filtering, the IFS shows the best result, with an increase in the graph homophilic rate of **8.65%**. This demonstrates that our novel connectivity measure is highly effective at reducing over-smoothing and increasing the graph homophily. In contrast, all other embedding-agnostic measures showed significantly worse results, proving that our approach of leveraging the rate of change of the embeddings throughout the message-passing successfully improves the topological edge-filtering. When analyzing the rate of change in the homophilic rate during heterophilic

Table 6: Prediction result comparison of various methods on four datasets with varying density.

| Methods | Organ-S | | Organ-S (dense) | | Organ-C | | Organ-C (dense) | |
|---|---|---|---|---|---|---|---|---|
| | Accuracy | Specificity | Accuracy | Specificity | Accuracy | Specificity | Accuracy | Specificity |
| GCN (Kipf & Welling, 2016) | 59.25±0.40 | 95.92±0.04 | 57.80±0.41 | 95.78±0.04 | **77.25±0.29** | **97.72±0.03** | 74.65±0.62 | 97.46±0.06 |
| GCN + *rewiring(DC)* | 59.02±0.20 | 95.90±0.02 | **58.25±0.42** | **95.82±0.04** | 76.79±0.29 | 97.68±0.03 | 74.92±0.19 | 97.49±0.02 |
| GCN + *rewiring(EC)* | 58.96±0.44 | 95.90±0.04 | 57.94±0.43 | 95.79±0.04 | 77.24±0.25 | 97.72±0.02 | **75.17±0.31** | **97.51±0.03** |
| GCN + *rewiring(BC)* | OOT | OOT | OOT | OOT | OOT | OOT | OOT | OOT |
| GCN + *rewiring(CC)* | OOT | OOT | OOT | OOT | 77.10±0.21 | 97.71±0.02 | OOT | OOT |
| GCN + *rewiring(FC)* | **59.48±0.31** | **95.95±0.03** | OOT | OOT | OOT | OOT | OOT | OOT |
| GCN + *rewiring(RC)* | OOT | OOT | OOT | OOT | OOT | OOT | OOT | OOT |
| GCN + *rewiring(IFS)* | 59.11±0.50 | 95.91±0.05 | 57.46±0.44 | 95.75±0.04 | 76.74±0.15 | 97.67±0.01 | 73.80±0.32 | 97.38±0.03 |
| GIN (Xu et al., 2018a) | 61.12±0.48 | 96.11±0.05 | 61.47±0.58 | 96.15±0.06 | 77.44±0.55 | 97.74±0.05 | 77.68±2.20 | 97.77±0.22 |
| GIN + *rewiring(DC)* | **61.85±1.30** | **96.18±0.13** | **62.31±0.29** | **96.23±0.03** | 77.31±2.03 | 97.73±0.20 | **78.48±0.59** | **97.85±0.06** |
| GIN + *rewiring(EC)* | 61.12±0.23 | 96.11±0.02 | 61.90±0.67 | 96.18±0.07 | 78.10±0.54 | 97.81±0.05 | 78.06±0.67 | 97.80±0.07 |
| GIN + *rewiring(BC)* | OOT | OOT | OOT | OOT | OOT | OOT | OOT | OOT |
| GIN + *rewiring(CC)* | OOT | OOT | OOT | OOT | OOT | OOT | OOT | OOT |
| GIN + *rewiring(FC)* | 61.60±1.23 | 96.16±0.12 | OOT | OOT | **78.94±0.91** | **97.89±0.09** | OOT | OOT |
| GIN + *rewiring(RC)* | OOT | OOT | OOT | OOT | OOT | OOT | OOT | OOT |
| GIN + *rewiring(IFS)* | 61.52±0.34 | 96.15±0.03 | 62.21±0.71 | 96.22±0.07 | 78.22±1.21 | 97.82±0.12 | 77.52±0.91 | 97.75±0.09 |
| GAT (Velickovic et al., 2017) | 52.57±0.18 | 95.25±0.02 | OOM | OOM | 69.27±0.63 | 96.93±0.06 | OOM | OOM |
| GAT + *rewiring(DC)* | 53.24±0.26 | 95.32±0.03 | OOM | OOM | 69.38±1.09 | 96.94±0.11 | OOM | OOM |
| GAT + *rewiring(EC)* | 53.03±1.32 | 95.30±0.13 | OOM | OOM | **69.84±0.05** | **96.98±0.00** | OOM | OOM |
| GAT + *rewiring(BC)* | OOT | OOT | OOT | OOT | OOT | OOT | OOT | OOT |
| GAT + *rewiring(CC)* | OOT | OOT | OOT | OOT | OOT | OOT | OOT | OOT |
| GAT + *rewiring(FC)* | **53.39±0.16** | **95.34±0.02** | OOT | OOT | 68.88±0.86 | 96.89±0.09 | OOT | OOT |
| GAT + *rewiring(RC)* | OOT | OOT | OOT | OOT | OOT | OOT | OOT | OOT |
| GAT + *rewiring(IFS)* | 51.49±1.24 | 95.15±0.12 | OOM | OOM | 68.52±2.12 | 96.85±0.21 | OOM | OOM |
| DeltaGNN - *control* | 62.62±0.08 | 96.26±0.01 | 62.34±0.43 | 96.23±0.04 | **81.44±0.16** | **98.14±0.01** | 80.53±0.68 | 98.05±0.07 |
| DeltaGNN - *control + DC* | 62.63±0.28 | 96.26±0.03 | **63.10±0.26** | **96.31±0.03** | 80.59±1.00 | 98.06±0.10 | **80.55±1.00** | **98.05±0.10** |
| DeltaGNN - *control + EC* | **62.90±0.40** | **96.29±0.04** | 62.87±0.10 | 96.28±0.01 | 79.70±1.35 | 97.97±0.13 | 79.20±0.84 | 97.91±0.08 |
| DeltaGNN - *control + BC* | OOT | OOT | OOT | OOT | OOT | OOT | OOT | OOT |
| DeltaGNN - *control + CC* | OOT | OOT | OOT | OOT | OOT | OOT | OOT | OOT |
| DeltaGNN - *control + FC* | 62.69±0.13 | 96.27±0.01 | OOT | OOT | 80.59±1.10 | 98.06±0.11 | OOT | OOT |
| DeltaGNN - *control + RC* | OOT | OOT | OOT | OOT | OOT | OOT | OOT | OOT |
| DeltaGNN *constant* | 62.69±0.76 | 96.27±0.08 | 62.54±0.46 | 96.25±0.05 | 79.71±0.30 | 97.97±0.03 | 79.13±0.24 | 97.91±0.02 |
| DeltaGNN *linear* | 62.09±0.37 | 96.21±0.04 | 62.04±0.23 | 96.20±0.02 | 79.23±0.33 | 97.91±0.03 | 79.26±0.27 | 97.92±0.03 |

Notes: Results are **bolded** if they are the best variation for a certain model. Green-shaded cells highlight the best three results overall. Results better than their counterparts have a more intense shade of green. OOT indicates an out-of-time error which is generated when the computation of the connectivity measure overruns 30 minutes. OOM indicates an out-of-memory error.

Table 7: Comparison of computational resources on datasets of increasing size and density.

| Methods | CORA | | | CiteSeer | | | PubMed | | | Organ-S | | | Organ-S (dense) | | |
|---|---|---|---|---|---|---|---|---|---|---|---|---|---|---|---|
| | GPU Memory | Preproc. Time | Epoch Time | GPU Memory | Preproc. Time | Epoch Time | GPU Memory | Preproc. Time | Epoch Time | GPU Memory | Preproc. Time | Epoch Time | GPU Memory | Preproc. Time | Epoch Time |
| GCN (Kipf & Welling, 2016) | **161.81** | **0.00** | 0.0014 | **267.60** | **0.00** | 0.0014 | **779.53** | **0.00** | 0.0015 | **641.31** | **0.00** | 0.0016 | **655.54** | **0.00** | 0.0017 |
| GCN + *rewiring(DC)* | **161.81** | 12.7536 | 0.0439 | **267.60** | 16.0516 | **0.0549** | **779.53** | 123.0887 | 0.8221 | **641.31** | 163.7771 | 0.8205 | **655.54** | 167.4163 | **0.8388** |
| GCN + *rewiring(EC)* | **161.81** | 12.7966 | 0.0440 | **267.60** | 16.0762 | 0.0014 | **779.53** | 128.6635 | 0.8592 | **641.31** | 165.3687 | 0.8284 | **655.54** | 168.4002 | 0.8437 |
| GCN + *rewiring(BC)* | **161.81** | 36.6763 | 0.1236 | **267.60** | 38.3571 | 0.1326 | OOT | OOT | OOT | OOT | OOT | OOT | OOT | OOT | OOT |
| GCN + *rewiring(CC)* | **161.81** | 16.5316 | 0.0565 | **267.60** | 18.8500 | 0.0642 | OOT | OOT | OOT | OOT | OOT | OOT | OOT | OOT | OOT |
| GCN + *rewiring(FC)* | **161.81** | 13.2862 | 0.0457 | **267.60** | 16.0841 | 0.0550 | **779.53** | 123.0187 | 0.8216 | **641.31** | 544.1153 | 2.7222 | OOT | OOT | OOT |
| GCN + *rewiring(RC)* | **161.81** | 14.2089 | 0.0488 | **267.60** | 16.7696 | 0.0573 | OOT | OOT | OOT | OOT | OOT | OOT | OOT | OOT | OOT |
| GCN + *rewiring(IFS)* | **161.81** | 13.5477 | 0.0465 | **267.60** | 17.3902 | 0.0594 | **779.53** | 137.0603 | 0.9152 | **641.31** | 186.4637 | 0.9339 | **655.54** | 188.1741 | 0.9426 |
| GIN (Xu et al., 2018a) | 401.25 | **0.00** | 0.0019 | 507.72 | **0.00** | 0.0020 | 1078.47 | **0.00** | 0.0021 | 867.77 | **0.00** | 0.0022 | 876.75 | **0.00** | 0.0024 |
| GIN + *rewiring(DC)* | 401.25 | 12.7536 | 0.0444 | 507.72 | 16.0516 | 0.0555 | 1078.47 | 123.0887 | 0.8227 | 867.77 | 163.7771 | 0.8211 | 876.75 | 167.4163 | 0.8395 |
| GIN + *rewiring(EC)* | 401.25 | 12.7966 | 0.0445 | 507.72 | 16.0762 | 0.0020 | 1078.47 | 128.6635 | 0.8598 | 867.77 | 165.3687 | 0.8290 | 876.75 | 168.4002 | 0.8444 |
| GIN + *rewiring(BC)* | 401.25 | 36.6763 | 0.1241 | 507.72 | 38.3571 | 0.1332 | OOT | OOT | OOT | OOT | OOT | OOT | OOT | OOT | OOT |
| GIN + *rewiring(CC)* | 401.25 | 16.5316 | 0.0570 | 507.72 | 18.8500 | 0.0648 | OOT | OOT | OOT | OOT | OOT | OOT | OOT | OOT | OOT |
| GIN + *rewiring(FC)* | 401.25 | 13.2862 | 0.0462 | 507.72 | 16.0841 | 0.0556 | 1078.47 | 123.0187 | 0.8222 | 867.77 | 544.1153 | 2.7228 | OOT | OOT | OOT |
| GIN + *rewiring(RC)* | 401.25 | 14.2089 | 0.0493 | 507.72 | 16.7696 | 0.0579 | OOT | OOT | OOT | OOT | OOT | OOT | OOT | OOT | OOT |
| GIN + *rewiring(IFS)* | 401.25 | 13.5477 | 0.0470 | 507.72 | 17.3902 | 0.0600 | 1078.47 | 137.0603 | 0.9158 | 867.77 | 186.4637 | 0.9345 | 876.75 | 188.1741 | 0.9432 |
| GAT (Velickovic et al., 2017) | 242.09 | **0.00** | 0.0217 | 336.31 | **0.00** | 0.0394 | 2271.71 | **0.00** | 0.0297 | 2184.78 | **0.00** | 0.1042 | OOM | OOM | OOM |
| GAT + *rewiring(DC)* | 242.09 | 12.7536 | 0.0642 | 336.31 | 16.0516 | 0.0929 | 2271.71 | 123.0887 | 0.8503 | 2184.78 | 163.7771 | 0.9231 | OOM | OOM | OOM |
| GAT + *rewiring(EC)* | 242.09 | 12.7966 | 0.0643 | 336.31 | 16.0762 | 0.0930 | 2271.71 | 128.6635 | 0.8874 | 2184.78 | 165.3687 | 0.9310 | OOM | OOM | OOM |
| GAT + *rewiring(BC)* | 242.09 | 36.6763 | 0.1439 | 336.31 | 38.3571 | 0.1706 | OOT | OOT | OOT | OOT | OOT | OOT | OOT | OOT | OOT |
| GAT + *rewiring(CC)* | 242.09 | 16.5316 | 0.0768 | 336.31 | 18.8500 | 0.1022 | OOT | OOT | OOT | OOT | OOT | OOT | OOT | OOT | OOT |
| GAT + *rewiring(FC)* | 242.09 | 13.2862 | 0.0660 | 336.31 | 16.0841 | 0.0930 | 2271.71 | 123.0187 | 0.8498 | 2184.78 | 544.1153 | 2.8248 | OOT | OOT | OOT |
| GAT + *rewiring(RC)* | 242.09 | 14.2089 | 0.0691 | 336.31 | 16.7696 | 0.0953 | OOT | OOT | OOT | OOT | OOT | OOT | OOT | OOT | OOT |
| GAT + *rewiring(IFS)* | 242.09 | 13.5477 | 0.0668 | 336.31 | 17.3902 | 0.0974 | 2271.71 | 137.0603 | 0.9434 | 2184.78 | 186.4637 | 1.0365 | OOM | OOM | OOM |
| DeltaGNN - *control* | 539.10 | 15.4632 | 0.0540 | 732.90 | 18.3549 | 0.0639 | 1891.55 | 142.3362 | 0.9518 | 1512.34 | 213.6147 | 1.0711 | 1520.61 | 245.9796 | 1.643 |
| DeltaGNN - *control + DC* | 539.10 | 14.4934 | 0.0508 | 732.90 | 18.3611 | 0.0639 | 1891.55 | 131.6359 | 0.8805 | 1512.34 | 188.6704 | 0.9463 | 1520.61 | 214.6011 | 1.4338 |
| DeltaGNN - *control + EC* | 539.10 | 14.7367 | 0.0516 | 732.90 | 17.7273 | 0.0618 | 1891.55 | 131.9083 | 0.8823 | 1512.34 | 197.7688 | 0.9918 | 1520.61 | 226.2949 | 1.5117 |
| DeltaGNN - *control + BC* | 539.10 | 38.9384 | 0.1323 | 732.90 | 41.0206 | 0.1394 | OOT | OOT | OOT | OOT | OOT | OOT | OOT | OOT | OOT |
| DeltaGNN - *control + CC* | 539.10 | 18.5498 | 0.0643 | 732.90 | 20.8089 | 0.0720 | OOT | OOT | OOT | OOT | OOT | OOT | OOT | OOT | OOT |
| DeltaGNN - *control + FC* | 539.10 | 15.1687 | 0.0531 | 732.90 | 18.0004 | 0.0627 | 1891.55 | 135.6264 | 0.9071 | 1512.34 | 555.8835 | 2.7824 | OOT | OOT | OOT |
| DeltaGNN - *control + RC* | 539.10 | 15.4131 | 0.0539 | 732.90 | 18.4731 | 0.0643 | OOT | OOT | OOT | OOT | OOT | OOT | OOT | OOT | OOT |
| DeltaGNN *constant* | 539.14 | **0.00** | **0.0372** | 732.87 | **0.00** | 0.0793 | 1897.26 | **0.00** | **0.3860** | 1540.95 | **0.00** | **0.7528** | 1570.87 | **0.00** | 1.2028 |
| DeltaGNN *linear* | 539.14 | **0.00** | **0.0372** | 732.87 | **0.00** | 0.0793 | 1897.26 | **0.00** | **0.3860** | 1540.95 | **0.00** | **0.7528** | 1570.87 | **0.00** | 1.2028 |

Notes: Results are **bolded** if they are the best result for a certain dataset. OOT indicate an out-of-time error which is generated when the computation of the connectivity measure overruns 30 minutes. OOM indicates an out-of-memory error. The GPU memory is in MB, the epoch and preprocessing times are in seconds.

graph condensation, we observe that the IFS yields the lowest variation. The highest rate of change is observed with the Ollivier-Ricci curvature, as illustrated in Figure 8 by the substantial decrease in the number of heterophilic nodes with a homophilic rate close to 0.1 during curvature-based edge filtering. Nevertheless, we can also observe that the performance of betweenness and closeness centrality during heterophilic edge condensation differs substantially. Given their theoretical similarity, this suggests that the role of the connectivity measure in heterophilic graph condensation might be marginal. Therefore, further experiments are needed to determine whether this disparity could impact the model performance.

Table 8: Variations in the homophilic rate during topological edge-filtering and heterophilic graph condensation using different connectivity measures. Results are **bolded** if they represent the best performance overall. Results that are better than their counterparts are shaded in progressively darker green.

| Methods | | | CORA | | |
| | Original Graph | Homophilic Filtered Graph | Δ Homophilic Rate | Heterophilic Condensed Graph | Δ Homophilic Rate |
| --- | --- | --- | --- | --- | --- |
| Degree Centrality | 0.6676 | 0.7115 | +6.72% | 0.981 | -85.28% |
| Eigenvector Centrality | 0.6676 | 0.6937 | +4.05% | 0.1122 | -83.17% |
| Betweenness Centrality | 0.6676 | 0.7057 | +5.85% | 0.905 | -86.42% |
| Closeness Centrality | 0.6676 | 0.7043 | +5.64% | 0.1244 | -81.34% |
| Forman-Ricci Curvature | 0.6676 | 0.7094 | +6.40% | 0.955 | -85.67% |
| Ollivier-Ricci Curvature | 0.6676 | 0.7112 | +6.67% | 0.767 | **-88.49%** |
| Information Flow Score | 0.6676 | 0.7244 | **+8.65%** | 0.1560 | -76.60% |

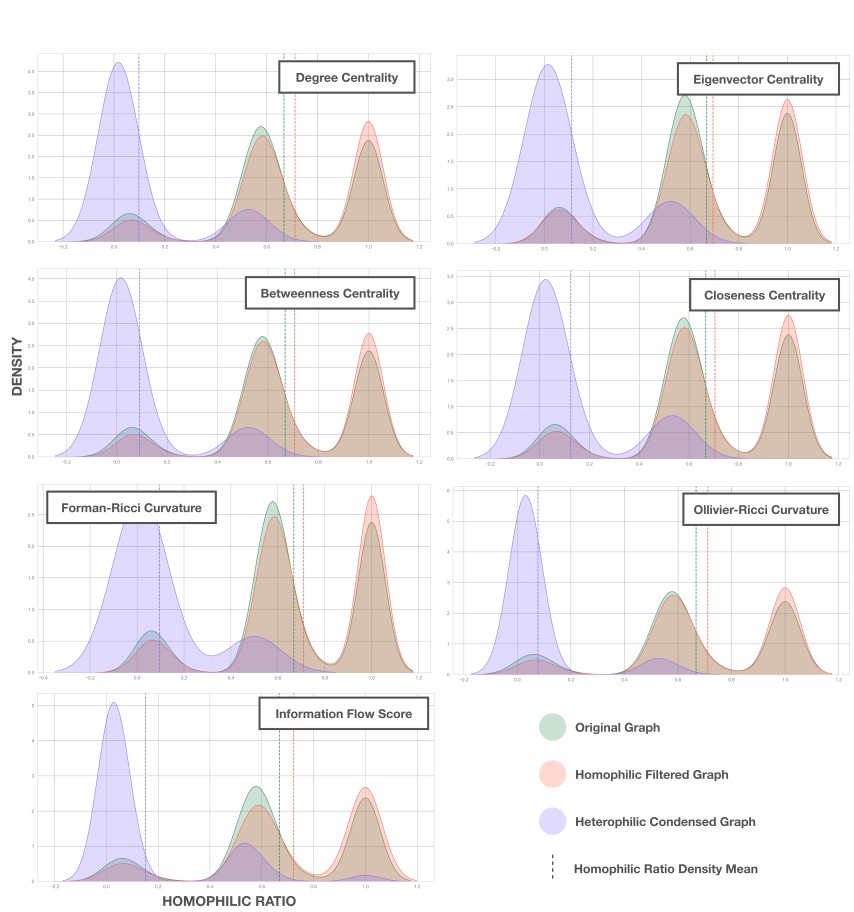

Figure 8: Homophilic ratio density distribution shifts during topological edge-filtering and heterophilic graph condensation using seven distinct connectivity measures. The dashed lines indicate the mean density for each corresponding measure and density.

