# OpenReview forum: "DeltaGNN: Graph Neural Network with Information Flow Control"
_ICLR.cc/2025/Conference — Submitted to ICLR 2025_

### Official Review · Reviewer_esWr · 2024-10-26

**Soundness:** 2
**Presentation:** 2
**Contribution:** 2
**Rating:** 3
**Confidence:** 4

**Summary:**

The paper introduces a score based on node features aggregated in a GNN layer, that aims at capturing the likelihood of a node to be responsible for oversmoothing and oversquashing. By leveraging such score in a graph-filtering pipeline, the authors propose a framework to alter the graph connectivity within a GNN scheme.

**Strengths:**

I think that it is valuable addressing issues like oversquashing and oversmoothing simultaneously, rather than studying them in isolation and independently of one another. I also liked the idea of leveraging "moments" from the feature distribution at different layers to guide the graph-filtering process.

**Weaknesses:**

There are important aspects of the submission that require reworking.

**Message and Presentation**

- In the introduction, there is often some ambiguity in the way you mention oversmoothing and oversquashing, as if they were interchangeable concepts. This is not the case, and should be emphasized. Oversmoothing is a problem that occurs for *some* GNNs and is independent of the topology (as a phenomenon, not how quickly that occurs) and is somewhat orthogonal to long-range interactions since in the limit of many layers, node features become indistinguishable irrespectively of their distance. Oversquashing instead, is an issue that occurs for *all* 1-hop GNNs and is very much dependent on the topology (namely, their commute time) and hence affects long-range interactions, independent of the depth or the ability to capture local interactions.
- Even more significantly, you keep overlapping the issue of oversmoothing with that of heterohily (for example Line 110, Line 121, Line 161 but this notion is repeated throughout the paper). This is wrong. While Definition 2.1 accounts for the labels, this to me represents more of a choice, as oversmoothing is the convergence of node features to the same representation over a connected component of the graph. As such, it is actually simply caused by low-frequencies dominating over high-frequencies in the graph spectrum. In fact, it can be mitigated or avoided by relying on architectures that do not operate via low-pass filters. I suspect that what you are implying here, is that oversmoothing becomes more of an issue in the presence of heterophily, as nodes with different labels become indistinguishable, but *this is a consequence of and not the cause of oversmoothing and should be rectified*.

- Quite a few citations are missing in the related work, for example regarding rewiring [1,2,3] but also Graph-Transformers..
- The presentation of the framework is a little contrived (see my questions below). Also, while you try to distinguish yourself from graph-rewiring algorithms, your approach removes edges, and this is a key part of it. For this reason, I think it is a little misleading to distinguish yourself from graph rewiring techniques. You should be more specific, and mention that the rewiring is adaptive and based on GNN layer outputs more than topological connectivity measures.

**Theory**

- I am a little confused by Lemma 1. To me, the homophily of a node only depends on the label information and the topology and has nothing to do with the architecture being used and/or the features. This indeed seems to be reflected also in your Definition 2.2 where I am reading that $\Phi$ can be taken to be the ground-truth label assignment. However, it seems that in Lemma 1 you are deriving the homophily of a node based on what can be mapped/separated from the node features, i.e. it has more to do with distinguishability from node features. If so, this should be clearly emphasized. As such, I would not really talk about homophily but node features separability.

- I don’t think that Lemma 2 is an actual Lemma since your proof is essentially a discussion based on  the results of Nguyen et al. You should remove the statement and replace it with a discussion based on what you have in the appendix. As it stands, I find it confusing and indeed informal, to a point that this is not a mathematical statement.

- In light of my comments regarding Lemma 2, I don’t think that your score definition is that well motivated. More precisely, I can see why the denominator makes sense in relation to oversmoothing, since it measures node features separability after rounds of message passing (and *not* homophily), but I struggle to see how the numerator relates to oversquashing. You should expand on the “proof” from Lemma 2, which is not really a proof, to better motivate this score.

**Experiments**

Evaluation is not  convincing. On all the benchmarks you used, it is highly debatable that long-range interactions are present at any level. In fact, I believe majority of people would argue that LRIs are not present on Cora, Pubmed, etc.. Additionally, datasets like Texas, Wisconsin, etc are known to have several issues and the community has proposed alternative options. I personally struggle to accept claims of “state of the art improvements by 1 %” on the likes of Cora and Pubmed this day. Graphs like Cornell, Texas and Wisconsin are also extremely small and super sensitive to tuning. The paper overall proposed a methodology, and as such, should be thoroughly tested on more relevant benchmarks.

[1]: Mitchell Black, Zhengchao Wan, Amir Nayyeri, and Yusu Wang. Understanding oversquashing in
gnns through the lens of effective resistance, ICML23.

[2]: Adrián Arnaiz-Rodríguez, Ahmed Begga, Francisco Escolano, and Nuria Oliver. DiffWire: Inductive
Graph Rewiring via the Lovász Bound, LOG 2022.

[3]: Kedar Karhadkar, Pradeep Kr Banerjee, and Guido Montúfar. Fosr: First-order spectral rewiring for
addressing oversquashing in gnns, 2022.

**Questions:**

- Equation (1) is not the most general way of writing a 1-hop GNN aggregation, as there is no residual term. Namely, one would typically expect $\phi$ to take two arguments i.e. $(\mathbf{X}_u^t, \bigoplus...)$
- Line 159: The expression “embedding agnostic” is a little vague to me, so perhaps you can specify a little more clearly what you are implying here.
- Line 285: What is a “homophilic GNN”?
- The paragraph 283-291 uses too many vague words and is all but clear. For example, line 288-289, what would an “heterophilic graph condensation” be?
- Line 330–331: How can removing edges that are bottlenecks necessarily reduce oversquashing? What if now you have disconnected components? This process can only work if one identifies correctly node labels, but this is something that your algorithm in general cannot know in advance.

---

> ### Author Response · Authors · 2024-11-27
>
> We sincerely thank the reviewer for their thoughtful and detailed feedback, which includes many important points that we are pleased to address. Below, we provide detailed responses to your comments.
>
> *Message and Presentation*
>
> Over-smoothing and over-squashing are indeed distinct concepts. Both relate to long-range interactions, and both phenomena prevent deep GNNs from effectively capturing LRIs, thereby increasing the difficulty of achieving better node classification accuracy in domains where distant node relationships matter. While over-smoothing causes feature indistinguishability after numerous consecutive aggregations, over-squashing occurs when substantial information collapses into a fixed-size feature vector due to graph bottlenecks. In our work, we highlight the different natures of these two phenomena: over-squashing is topological, whereas over-smoothing happens in deep GNNs, and the speed at which over-smoothing occurs is related to heterophily. Recent works have shown promising results in addressing over-smoothing and heterophily simultaneously by introducing novel node-level metrics [1]. Similarly, we analyze the relationship between these two phenomena to propose a measure that can be leveraged to tackle both.
> Since graph heterophily results in faster feature over-smoothing as the number of layers increases (this occurs because representations are drawn toward the mean feature vector of dissimilar classes, reducing feature separability and further exacerbating over-smoothing at deeper layers), we propose a novel approach to increase the homophily of a graph through edge-filtering (see Appendices D and G). This slows down feature indistinguishability and indirectly alleviates the effects of over-smoothing. We have updated the manuscript to clarify the relationship between over-smoothing and heterophily, including appropriate references. In the previous version of the manuscript, we incorrectly suggested a causal relationship between over-smoothing and heterophily; we thank the reviewer for pointing this out.
> There are many ways to measure over-smoothing within a graph. While the Dirichlet energy was initially proposed as a viable measure of over-smoothing, numerous approaches now exist [2]. In our work, we aimed to define over-smoothing in the context of node classification tasks while highlighting its consequences on the feature distinguishability of different labels. As this is not the standard formalization of over-smoothing, the definition in the paper has been rectified.
> We have included additional citations in the “Related Work” section regarding rewiring algorithms and graph transformers.
> Our work is strongly inspired by graph-rewiring algorithms, as we aim to propose a more adaptive and integrated solution that leverages topological edge-rewiring while also using layer outputs to learn how to improve model performance. In the manuscript, we highlight the differences between our approach and recent edge-filtering methods. The crucial idea behind our work is to consider embeddings (and their momentum) to understand the graph structure and homophily. In this sense, our approach is not embedding-agnostic, which strongly differs from most popular approaches in the literature [3,4,5]. Consequently, our approach is the first to propose a truly scalable (with linear time complexity concerning the number of nodes) measure that simultaneously addresses over-smoothing and over-squashing.
>
> *References:*
>
> [1] Yan, Y., Hashemi, M., Swersky, K., Yang, Y. and Koutra, D., 2022, November. Two sides of the same coin: Heterophily and oversmoothing in graph convolutional neural networks. In 2022 IEEE International Conference on Data Mining (ICDM) (pp. 1287-1292). IEEE.
>
> [2] Rusch, T.K., Bronstein, M.M. and Mishra, S., 2023. A survey on oversmoothing in graph neural networks. arXiv preprint arXiv:2303.10993.
>
> [3] Karhadkar, K., Banerjee, P.K. and Montúfar, G., 2022. FoSR: First-order spectral rewiring for addressing oversquashing in GNNs. arXiv preprint arXiv:2210.11790.
>
> [4] Nguyen, K., Hieu, N.M., Nguyen, V.D., Ho, N., Osher, S. and Nguyen, T.M., 2023, July. Revisiting over-smoothing and over-squashing using ollivier-ricci curvature. In International Conference on Machine Learning (pp. 25956-25979). PMLR.
>
> [5] Fesser, L. and Weber, M., 2024, April. Mitigating over-smoothing and over-squashing using augmentations of Forman-Ricci curvature. In Learning on Graphs Conference (pp. 19-1). PMLR.

---

> ### Author Response · Authors · 2024-11-27
>
> *Theory*
>
> Lemmas 1 and 2 are important theoretical results that together motivate the definition of our novel Information Flow Score (IFS). Additional details are provided in Appendix D. Lemma 1 highlights the relationship between $\overline{\Delta_u}$ and graph homophily, demonstrating that high values of $\overline{\Delta_u}$ are associated with low node homophily (an expanded version of the proof is available in Appendix B). To offer the reviewer a more intuitive perspective that motivates these findings, we observe that when a node experiences high values of $\overline{\Delta_u}$, this phenomenon can be linked to the ratio of neighboring nodes with dissimilar features. Under certain assumptions about the distance between the subspaces of the feature space associated with different node classes (assumptions on feature separability), this observation implies that the homophilic ratio of a node must be below a specific threshold.
> This theoretical result is further supported by our numerical experiment in Appendix G, where we directly compare several topological measures (including our novel Information Flow Score) and analyze changes in the graph's homophilic ratio during edge-filtering in the CORA dataset. The IFS achieves the highest increase in the homophilic ratio, with the results summarized below:
>
> | **Methods**               | **Original Graph** | **Homophilic Filtered Graph** | **Δ Homophilic Rate** |
> |----------------------------|--------------------|-------------------------------|------------------------|
> | Degree Centrality          | 0.6676            | 0.7115                        | +6.72%                |
> | Eigenvector Centrality     | 0.6676            | 0.6937                        | +4.05%                |
> | Betweenness Centrality     | 0.6676            | 0.7057                        | +5.85%                |
> | Closeness Centrality       | 0.6676            | 0.7043                        | +5.64%                |
> | Forman-Ricci Curvature     | 0.6676            | 0.7094                        | +6.40%                |
> | Ollivier-Ricci Curvature   | 0.6676            | 0.7112                        | +6.67%                |
> | Information Flow Score     | 0.6676            | 0.7244                        | **+8.65%**            |
>
>
> On the other hand, Lemma 2 investigates the relationship between $\mathbb{V}_t[\Delta^2_u]$ and graph connectivity, demonstrating that high values of $\mathbb{V}_t[\Delta^2_u]$ are associated with graph bottlenecks. Specifically, high $\mathbb{V}_t[\Delta^2_u]$ values correspond to areas of the graph that quickly converge to fixed feature values, indicating that the node $u$ is highly connected within the graph. As a result, graph bottlenecks coincide with very low values of $\mathbb{V}_t[\Delta^2_u]$.
> We informally prove Lemma 2 by building on the results of Nguyen et al., reducing the need for abstruse and unnecessary mathematical notations to improve clarity (an expanded version of the proof is included in Appendix C). We hope these intuitions help clarify the roles of Lemmas 1 and 2 in our work. We also suggest that the reviewer refer to Appendix D.2, where we include a numerical experiment that provides additional graphical intuition.

---

> ### Author Response · Authors · 2024-11-27
>
> *Experiments*
>
> We appreciate the reviewer’s suggestion to consider more challenging heterophilic datasets. Unfortunately, due to time constraints, we are unable to include such experiments at this time. However, we would like to emphasize that we have already demonstrated that our DeltaGNN consistently delivers better performance on graph datasets with varying sizes, topologies, and levels of homophily. Specifically, we showed a reproducible and consistent reduction of the average epoch time by over 30% when using DeltaGNN, proving its scalability across datasets of varying sizes and edge densities. In particular, we benchmarked our model on 10 datasets containing up to 23,660 nodes and 2,809,204 edges. Please refer to Appendix D.2 for additional experiments on size-varying and density-varying graphs.
> While we agree that some of the benchmarked datasets have been contested in the literature due to their limited graph sizes, we also note that they remain the most widely benchmarked datasets for research related to graph rewiring and long-range interactions. This makes them a fair platform for evaluating our novel methodology.
> For instance, the works [3,6] you referenced were primarily benchmarked on the “TuDatasets” suite. We have also previously benchmarked our models on datasets such as MUTAG, ENZYMES, COLLAB, REDDIT-BINARY, IMDB-BINARY, and PROTEINS. However, we chose not to include these results due to the high variance in performance and questionable reliability of these datasets. This issue has also been noted by [6] in a recent study, where they state: “First, many of the datasets contain few graphs. For instance, MUTAG contains 188 graphs, meaning that only 18 graphs are part of the test set. Further, REDDIT-BINARY, IMDB-BINARY, and COLLAB do not have node features and are augmented with constant feature vectors. Consequently, it is not immediately clear how much long-range interactions play a role in these tasks, or, in fact, how to even define over-squashing on graphs without (meaningful) features.”
> For these reasons, we would appreciate clarification regarding which datasets you consider “relevant benchmarks.” This feedback will help us consolidate and expand the variety and reliability of our experimental evaluations in future research. However, for fairness, we believe this request should not detract from acknowledging that our work adhered to the same experimental protocols as most studies in this research area.
>
> *Questions:*
>
> 1.We updated the equation by extending the definition of the neighborhood to include the node u itself (similarly to related works [4]).
>
> 2.Already addressed in previous points.
>
> 3.Throughout the paper, we use the term "heterophilic GNN" to refer to a GNN architecture that implements an aggregation paradigm designed to handle heterophilic neighborhoods. In contrast, the term "homophilic GNN" refers to a GNN architecture that implements a standard aggregation paradigm (e.g., mean, sum) optimized for homophilic neighborhoods. This distinction between homophilic and heterophilic aggregation is crucial to avoid the misconception that DeltaGNN employs the same aggregation paradigm for processing both homophilic and heterophilic graphs (see Figure 3).
>
> 4/5. “Heterophilic Graph Condensation” is a term we coined to describe the specific step in the DeltaGNN pipeline where we construct a condensed graph that primarily includes heterophilic edges, thereby reintroducing long-range node dependencies. This step is crucial because, as you observed, removing graph bottlenecks may lead to disconnected components and the loss of long-range node dependencies. Additional technical details can be found in lines 362–..., and an illustrative graphic of this process is provided in Figure 3. Finally, we included a numerical experiment in Appendix G where we measure the graph's homophily after edge-filtering and after heterophilic graph condensation.
>
> We sincerely thank the reviewer once again for the excellent questions and hope that we have addressed any doubts in the new version of the paper. We would like to emphasize that responding to these points was both challenging—given that they touch on core aspects of our work—and beneficial, as they helped us better articulate our contributions. For these reasons, we are truly grateful for the reviewer’s interest in our work. We look forward to further engagement and hope this encourages an improved score.
>
>
> *References:*
>
> [3] Karhadkar, K., Banerjee, P.K. and Montúfar, G., 2022. FoSR: First-order spectral rewiring for addressing oversquashing in GNNs. arXiv preprint arXiv:2210.11790.
>
> [4] Nguyen, K., Hieu, N.M., Nguyen, V.D., Ho, N., Osher, S. and Nguyen, T.M., 2023, July. Revisiting over-smoothing and over-squashing using ollivier-ricci curvature. In International Conference on Machine Learning. PMLR.
>
> [6]: Adrián Arnaiz-Rodríguez, Ahmed Begga, Francisco Escolano, and Nuria Oliver. DiffWire: Inductive Graph Rewiring via the Lovász Bound, LOG 2022.

---

### Official Review · Reviewer_6W5k · 2024-11-02

**Soundness:** 2
**Presentation:** 2
**Contribution:** 2
**Rating:** 5
**Confidence:** 4

**Summary:**

The paper  introduces a mechanism to mitigate over-smoothing and over-squashing in Graph Neural Networks (GNNs) by implementing an "information flow control" strategy that utilizes an "information flow score." This approach allows for effective management of node embeddings across varied graph structures, demonstrating enhanced performance in large-scale graphs while maintaining computational efficiency.

**Strengths:**

originality: good
quality: medium
clarity: medium
significance: medium

**Weaknesses:**

1. "These long-range interactions (LRIs) are crucial for node classification tasks, as they help distinguish between different classes and improve classification accuracy" This is not true. For example, graph transformers are good at capturing long-range node dependencies. However, they perform poorly on node classification tasks, especially on heterophilic graphs [1]. It is found that distant information is not always useful, and the over-globalization can cause performance degradation of graph models [2].
2. "over-smoothing is not only a topological phenomenon but is primarily a consequence of graph heterophily." There is no causal relation between over-smoothing and heterophily. As stated in [3], over-smoothing only happens in deep GNNs, but not in shallow GNNs. Heterophily will cause performance degradation to all GNN models, not matter they are deep or shallow.

**Questions:**

1. the information flow score, which identifies graph bottlenecks and heterophilic node interactions,

2. In definition 1, the first Delta embeddings look like the "norm" of the high-pass filtered graph signal or the neighborhood diversification[4]. The second Delta embedding is a new and interesting one.

3. So how can lemma 1 and 2 offer insights into the graph’s homophily and topology? Explain with sentences.

4. How did you get equation (2)? Why "nodes with low values of this measure are likely to correspond to regions where over-smoothing and over-squashing occur"?

5. "The long-range dependencies are then learned via a GNN heterophilic aggregation." What is "heterophilic aggregation"? Do you mean aggregation from long-range nodes in different classes? Are such long-range dependency beneficial?

6. "This concept of homophily-based interaction-decoupling is crucial to prevent over-smoothing by avoiding using a standard GNN aggregation on heterophilic edges." The "decoupling" is indeed important, for example in [4], the authors use 3-channel architectures to address heterophily. But the objective is not to prevent over-smoothing, it is to improve node distinguishability [5]. A direct proof on why and how your proposed method can improve node distinguishability is recommended.

7. Missing comparison with some SOTA models on heterophilic graphs, e.g. [4,6,7]. More comparisons on the real challenging heterophilic datasets suggested in [3] are recommended.



[1] Müller L, Galkin M, Morris C, Rampášek L. Attending to Graph Transformers. Transactions on Machine Learning Research.

[2] Less is More: on the Over-Globalizing Problem in Graph Transformers. InForty-first International Conference on Machine Learning.

[3] The heterophilic graph learning handbook: Benchmarks, models, theoretical analysis, applications and challenges. arXiv preprint arXiv:2407.09618. 2024 Jul 12.

[4] Revisiting heterophily for graph neural networks. Advances in neural information processing systems. 2022 Dec 6;35:1362-75.

[5] When Do Graph Neural Networks Help with Node Classification? Investigating the Homophily Principle on Node Distinguishability. Advances in Neural Information Processing Systems. 2024 Feb 13;36.

[6] Simplifying approach to node classification in graph neural networks[J]. Journal of Computational Science, 2022, 62: 101695.

[7] Diverse message passing for attribute with heterophily[J]. Advances in Neural Information Processing Systems, 2021, 34: 4751-4763.

---

> ### Author Response · Authors · 2024-11-25
>
> We appreciate that the reviewer found our work interesting and our ideas novel. Below, we address the questions and concerns raised, which we believe have strengthened our work.
>
>
> *(Weakness 1): “The role of long-range interactions”*
>
> We thank the reviewer for the compelling questions as they present important and interesting points to further discuss. **While it is true that some graph transformers perform poorly on node classification tasks involving large heterophilic graphs, this does not imply that LR interactions in such tasks are of marginal importance**. In fact, as it has been observed in [1]: *“... when applied to graphs with 10K or more nodes, the self-attention of graph transformers is subjected to much more noise, which makes learning meaningful long-range interactions difficult”*. In general, when processing large or heterophilic graphs, understanding global patterns and long-range dependencies is harder due to the higher amount of noise. Moreover, recent works on node classification tasks outline that positional/structural information (for example Laplacian Positional Encoding) may be enough to get SOTA performance [1]. Including positional/structural information in the node feature vectors is a way of encoding global and relative structural context to nodes in Graph Neural Networks, which is another way of tackling the long-range dependencies problem. Indeed, long-range interactions are useful to improve node classification accuracies but in some domains models may struggle to properly capture them.
>
>
> *(Weakness 2): “Over-smoothing and heterophily”*
>
> We appreciate the reviewer’sobservation and we would like to clarify the statement made in the manuscript. We concur that there is no causal relationship between over-smoothing and heterophily as these two problems are classically defined as independent phenomena. However, recent works have shown promising results trying to tackle over-smoothing and heterophily simultaneously by introducing novel node-level metrics [2]. Similarly, here, we are analysing the relationship between these two phenomena to propose a measure which can be leveraged to tackle **both**. Since **graph heterophily results in a faster features over-smoothing as the number of layers increases** (this happens because representations are drawn toward the mean feature vector of dissimilar classes, reducing feature separability and further exacerbating the over-smoothing problem at deeper layers), we propose a novel approach which increases the homophily of a graph through a edge-filtering (see Appendix D and G) slowing down feature indistinguishability and indirectly alleviating the effects of over-smoothing. We updated the manuscript relaxing the contested statement. Indeed, we are happy to further discuss this point if the reviewer finds it appropriate.
>
>
> *Questions:*
>
> *“First and Second Delta Embeddings”*
>
> We are glad to hear that the reviewer found our contribution original. The First and Second delta embeddings are indeed key innovative concepts in our work. The First delta embeddings $\Delta^t_u$ is a sequence which measures the aggregation velocity at every layer of the GNN. The concept of aggregation velocity is the physical interpretation of the rate of change of the node embeddings throughout the aggregation within a GNN layer. **The higher this number, the faster a node embedding moves within the feature space at a certain time t**. The Second delta embeddings $(\Delta^2)^t_u$ is defined as the differences of $\Delta^t_u$ over time. This object measures the aggregation acceleration of the feature vectors over time. **The higher this number, the faster a node embedding converges within the feature space at a certain time t**. To give a graphical intuition we included a numerical experiment on a small graph of CT scans of liver tumours extracted from a MedMNIST medical dataset [3], where we depict how the first differences of the embeddings $\Delta^t_u$ and second differences $(\Delta^2)^t_u$ vary throughout a 4 layer DeltaGNN architecture (see Appendix D.2).
>
> *References:*
>
> [1] Müller, L., Galkin, M., Morris, C. and Rampášek, L., 2023. Attending to graph transformers. arXiv preprint arXiv:2302.04181.
>
> [2] Yan, Y., Hashemi, M., Swersky, K., Yang, Y. and Koutra, D., 2022, November. Two sides of the same coin: Heterophily and oversmoothing in graph convolutional neural networks. In 2022 IEEE International Conference on Data Mining (ICDM) (pp. 1287-1292). IEEE.
>
> [3] Yang, J., Shi, R., Wei, D., Liu, Z., Zhao, L., Ke, B., Pfister, H. and Ni, B., 2023. Medmnist v2-a large-scale lightweight benchmark for 2d and 3d biomedical image classification. Scientific Data, 10(1), p.41.

---

> ### Author Response · Authors · 2024-11-25
>
> *“Lemma 1 and 2”*
>
> Lemma 1 and 2 are important theoretical results which motivate the definition of our novel Information Flow Score. Lemma 1 highlights the relationship between $\overline{\Delta_u}$ and the graph homophily, demonstrating that high values of $\overline{\Delta_u}$ are linked to low node homophily (a detailed proof is included in Appendix B). On the other hand, Lemma 2 investigates the relationship between $\mathbb{V}_t[\Delta^2_u]$ and the graph connectivity, demonstrating that high values of $\mathbb{V}_t[\Delta^2_u]$ are linked to graph bottlenecks (an informal proof is included in Appendix C). Here, we would like to give the reviewer a more intuitive observation which motivates these facts. When a node experiences high values of $\overline{\Delta_u}$ this phenomenon can be linked to the ratio of neighbouring nodes with dissimilar features. Under certain assumptions on the distance between the subspaces of the feature space associated with the node classes, this fact can be used to show that the homophilic ratio of a node must be lower than a certain threshold. Similarly, high values of $\mathbb{V}_t[\Delta^2_u]$ can be linked to areas of the graph converging quickly to fixed feature values, which shows that the node $u$ must be highly connected within the graph. As a result, graph bottlenecks must coincide with very low values of $\mathbb{V}_t[\Delta^2_u]$. We hope these simple intuitions are helpful to understand these mechanics. We also suggest the reviewer have a look at Appendix D.2 where we included a numerical experiment which may give an additional graphical intuition to this topic.
>
>
> *“Information Flow Score equation”*
>
> Equation 2 defines the Information Flow Score, our **novel measure** designed to **detect areas of the graph affected by over-smoothing and over-squashing**. From Lemma 1, we know that a high value of $\overline{\Delta_u}$ is likely associated with a low homophilic ratio, and from Lemma 2, a low $\mathbb{V}_t[\Delta^2_u]$ indicates proximity to an edge bottleneck. Consequently, the score is defined as a fraction involving these two terms. This ensures that nodes with low values of our measure are likely to correspond to regions where over-smoothing and over-squashing occur. Since, in some applications, we may prioritize detecting bottlenecks over heterophilic edges (or vice versa), we introduce two multipliers, $l$ and $m$, to adjust the weights of the mean and variance in the final score. In our case, we aim to detect both with equal priority, so we set “$m = 1$” and “$l = 1$”. To ensure that the score remains well-defined, regardless of the delta values; and to prevent it from exploding when the deltas are close to zero, we added 1 to both the numerator and denominator. An additional advantage of adding 1 is that isolated nodes will receive a score of one, which is relatively low. As a result, training the model to rewire the graph while maximizing the overall score will encourage the model to remove edges while still preserving sparsity. A detailed analysis of the implementation of the Information Flow Score can be found in Appendix D.
>
>
> *“Heterophilic aggregation in DeltaGNN”*
>
> Throughout the paper, we use the term "heterophilic aggregation" to refer to an aggregation method designed to handle highly heterophilic neighborhoods.. In contrast, the term “homophilic aggregation” refers to a standard aggregation paradigm (e.g., mean, sum) that primarily supports homophilic neighborhoods. This distinction between homophilic and heterophilic aggregation is crucial to avoid the misconception that DeltaGNN should employ the same aggregation paradigm for processing both homophilic and heterophilic graphs.

---

> ### Author Response · Authors · 2024-11-25
>
> *“Interaction-Decoupling in DeltaGNN”*
>
> We thank the reviewer for the insightful observation. Yes, it is correct. The proposed dual paradigm is introduced to improve node distinguishability by increasing the graph homophily and consequently slowing down the effects of over-smoothing (this last implication follows from Lemma 1). The effectiveness of this approach can also be appreciated by comparing the accuracy of DeltaGNN with its ablations (see Table 1). Moreover, Figure 8 (in Appendix G) depicts the rate of change of the graph homophily during the edge-rewiring (comparing our approach with six popular connectivity measures) showing superior performance of our model while consolidating our theoretical findings. (we summarize here our results)
>
> | **Methods**               | **Original Graph** | **Homophilic Filtered Graph** | **Δ Homophilic Rate** |
> |----------------------------|--------------------|-------------------------------|------------------------|
> | Degree Centrality          | 0.6676            | 0.7115                        | +6.72%                |
> | Eigenvector Centrality     | 0.6676            | 0.6937                        | +4.05%                |
> | Betweenness Centrality     | 0.6676            | 0.7057                        | +5.85%                |
> | Closeness Centrality       | 0.6676            | 0.7043                        | +5.64%                |
> | Forman-Ricci Curvature     | 0.6676            | 0.7094                        | +6.40%                |
> | Ollivier-Ricci Curvature   | 0.6676            | 0.7112                        | +6.67%                |
> | Information Flow Score     | 0.6676            | 0.7244                        | **+8.65%**            |
>
> *“Additional  comparisons and experiments”*
>
> We would like to point out that **the main point of our work is to show a novel graph rewiring-inspired approach** for **simultaneously alleviating over-smoothing and over-squashing**, proposing a novel connectivity measure called Information Flow Score and architecture named DeltaGNN. For this reason, we believe that focusing on rewiring baselines is fair (which is aligned with recent works on this topic [4]). To strengthen our comparison we also included existing state-of-the-art graph-transformer baselines and heterophily-based methods such as NAGphormer [5] and H2GCN [6], even though these approaches are architecturally very different from ours and also computationally more expensive. As a result, we struggle to see how including additional comparisons with SOTA methods on heterophilic graphs could further strengthen our contributions. Additionally, we appreciate the reviewer’s suggestion to consider more challenging heterophilic datasets. Unfortunately, due to time constraints, we are unable to add such experiments at this time. We have already shown that our Delta GNN produces consistently better performance on graph datasets with varying sizes, topologies, and homophily. Specifically, we showed a reproducible and consistent reduction of the average epoch time of >30% when using DeltaGNN, proving its scalability with datasets of varying sizes and edge densities. **Specifically, we benchmarked our model on 10 datasets with up to 23660 nodes and 2’809’204 edges** (which is still considerably higher than some ogb datasets, like ogb-arxiv which includes on average graphs with ~1,166,243 edges). Please refer to Appendix D.2 for additional experiments on size-varying and density-varying graphs.
> In future research, we will consider including such datasets to improve experimental variety. We are happy to further discuss this point if necessary
>
>
> We thank once again the reviewer for the excellent questions. We really appreciate the detailed feedback and numerous references. Discussing such topics was truly beneficial to better articulate our contributions and we are happy to further engage with the reviewer.
>
> *References:*
>
> [4] Topping, J., Di Giovanni, F., Chamberlain, B.P., Dong, X. and Bronstein, M.M., 2021. Understanding over-squashing and bottlenecks on graphs via curvature. arXiv preprint arXiv:2111.14522.
>
> [5] Jinsong Chen, Kaiyuan Gao, Gaichao Li, and Kun He. Nagphormer: A tokenized graph transformer for node classification in large graphs. arXiv preprint arXiv:2206.04910, 2022.
>
> [6] Jiong Zhu, Yujun Yan, Lingxiao Zhao, Mark Heimann, Leman Akoglu, and Danai Koutra. Beyond homophily in graph neural networks: Current limitations and effective designs. Advances in neural information processing systems, 33:7793–7804, 2020.

---

> > ### Comment · Reviewer_6W5k · 2024-12-01
> > **Thanks for the reply**
> >
> > Thanks for the reply from the authors. After going through the comments from other reviewers, I will keep my rating.

---

### Official Review · Reviewer_e676 · 2024-11-04

**Soundness:** 3
**Presentation:** 3
**Contribution:** 2
**Rating:** 5
**Confidence:** 4

**Summary:**

The paper identifies that Long-Range Interactions (LRIs) are crucial for node classification tasks. Standard GNNs struggle to capture these long range dependencies due to issues such as over-smoothing and over-squashing. To address these challenges, the authors propose information flow control, a graph rewiring mechanism. Further, the paper introduces DeltaGNN, which implements information flow control to capture both long- and short-range dependencies. The proposed method is validated on several graph datasets with varying levels of homophily and sizes.

**Strengths:**

* The paper is well-written and easy to follow.

* It introduces a novel connectivity measure, called the information flow score, which is supported by both theoretical analysis and empirical evidence.

* DeltaGNN demonstrates consistent improvements across various datasets, outperforming all baseline methods compared in the study.

**Weaknesses:**

* DeltaGNN is proposed as a scalable approach for detecting both long-range and short-range interactions. However, there are no large-scale experiments to validate this claim, as all experiments were conducted on small graphs. It would be beneficial if the authors could report results on larger homophilic datasets, such as ogbn-arXiv, as well as on large-scale non-homophilous graphs from [1].

* The related work section does not adequately situate the current research within the context of existing GNN work based on Graph Filters (e.g., [3, 4]).

* Lines 361-363 indicate that DeltaGNN is compared against state-of-the-art (SoTA) GNNs. However, GCN, GAT, and GIN are not the current SoTA for the chosen benchmarks. The authors should compare DeltaGNN with more recent GNNs (e.g., ACM-GCN+ / ACMII-GCN++ from [2]) to more accurately assess its effectiveness.

* It is unclear why MLP is not included as a baseline in Table 1. MLP has been shown to outperform on the three non-homophilous datasets (Texas, Wisconsin, Cornell) as reported in [4]. A comparison against graph filter-based methods, such as GPR-GNN [3] or PPGNN [4], would provide further insights into the performance of DeltaGNN.

---
[1] Large Scale Learning on Non-Homophilous Graphs: New Benchmarks and Strong Simple Methods, NeurIPS 2021

[2] Revisiting Heterophily For Graph Neural Networks, NeurIPS 2022

[3] Adaptive Universal Generalized PageRank Graph Neural Network, ICLR 2021

[4] A Piece-Wise Polynomial Filtering Approach for Graph Neural Networks, ECML 2022

**Questions:**

* From Table 7 in the appendix, DeltaGNN variants consume approximately 2-3 times more GPU memory than GCN on small graphs. Could the authors discuss whether this would lead to memory issues when applied to larger graphs?

* Did the authors evaluate DeltaGNN on more challenging heterophilic datasets, such as Squirrel or Chameleon [3]?

*  [Minor] Typo in Line 181: "$∆^t_u$ the first" $\rightarrow$ "$∆^t_u$ be the first."

---

> ### Author Response · Authors · 2024-11-25
>
> We appreciate that the reviewer found our work interesting and our ideas novel. Below, we address the questions and concerns raised, which we believe have strengthened our work.
>
> *(Weakness 1) “Large homophilic and heterophilic graphs”*
>
>
> We agree with the reviewer that additional experiments on larger datasets would be beneficial for further supporting our strong results on varying datasets with different disparate topologies. Unfortunately, due to our limited hardware capacity, as many ICLR research groups, we struggle to properly tune and fit GNN models in datasets with hundreds of thousands of nodes in a reasonable time frame. **For fairness, we believe that this request should not deter reviewers from acknowledging the merits of such works when they are evaluated on small-to-large scale graph data —but not very large ones such as ogb**. We have already shown that our Delta GNN produces consistently better performance on graph datasets with varying sizes, topologies, and homophily. Specifically, we showed a reproducible and consistent reduction of the average epoch time of >30% when using DeltaGNN, proving its scalability with datasets of varying sizes and edge densities. **Specifically, we benchmarked our model on 10 datasets with up to 23660 nodes and 2’809’204 edges** (which is still considerably higher than some ogb datasets, like ogb-arxiv which includes on average graphs with ~1,166,243 edges). Please refer to Appendix D.2 for additional experiments on size-varying and density-varying graphs.
>
>
> *(Weakness 2/3) “Graph filters and additional comparisons”*
>
>
> We would like to point out that **one of the key main contributions of our work is to show a novel graph rewiring-inspired approach for alleviating over-smoothing and over-squashing**, proposing a novel connectivity measure called Information Flow Score and a novel architecture named DeltaGNN. For this reason, we believe that focusing on rewiring baselines is fair (which is aligned with recent works on this topic [1]). To strengthen our comparison we also included existing state-of-the-art graph-transformer baselines and heterophily-based methods such as NAGphormer [2] and H2GCN [3], even though these approaches are architecturally very different from ours and also computationally more expensive. As a result, we struggle to see how including additional comparisons such as Graph Filters could be beneficial for our research. We are happy to further discuss this point if the reviewer finds it appropriate.
>
>
> *(Weakness 4) “MLP baseline”*
> We appreciate the reviewer’s observations and we agree that including a comparison against MLP could provide further insights into the performance of DeltaGNN. The manuscript has been updated and a table summarizing the performance disparity is reported below.
>
> | **Methods**                 | **CORA** | **Citeseer** | **PubMed** | **Cornell** | **Texas** | **Wisconsin** |
> |-----------------------------|------------------------------------|---------------------------------------|-------------------------------------|-------------------------------------|-----------------------------------|-------------------------------------|
> | MPL   | 70.32±2.68                        | 68.64±1.98                            | 86.46±0.35                         | 71.62±5.57                         | **77.83±5.24**                    | **82.15±6.93**                     |
> | DeltaGNN *(best version)*   | **87.29±0.52**                    | **79.90±0.79**                        | **89.98±0.24**                     | **75.67±1.91**                     | 74.05±3.08                        | 80.00±0.88                         |
>
> *References:*
>
> [1] Topping, J., Di Giovanni, F., Chamberlain, B.P., Dong, X. and Bronstein, M.M., 2021. Understanding over-squashing and bottlenecks on graphs via curvature. arXiv preprint arXiv:2111.14522.
>
> [2] Jinsong Chen, Kaiyuan Gao, Gaichao Li, and Kun He. Nagphormer: A tokenized graph transformer for node classification in large graphs. arXiv preprint arXiv:2206.04910, 2022.
>
> [3] Jiong Zhu, Yujun Yan, Lingxiao Zhao, Mark Heimann, Leman Akoglu, and Danai Koutra. Beyond homophily in graph neural networks: Current limitations and effective designs. Advances in neural information processing systems, 33:7793–7804, 2020.

---

> ### Author Response · Authors · 2024-11-25
>
> *(Question 1) “Memory consumption”*
>
> While DeltaGNN proved to be very efficient in terms of time complexity showing the best results (see Table 2), it also results in a higher memory footprint (on average x2.4 compared to a simple GCN) due to its dual architecture which inherently requires more learnable parameters. On the other hand, this additional complexity is independent of the graph size, this can be appreciated in Table 7, where the memory disparity between DeltaGNN and GCN remains the same also when benchmarking “Organ-S (dense)” which has 25’221 nodes and 2’494’750 edges. Additionally, we want to emphasize that, as demonstrated in the paper, our model remains more scalable in terms of memory consumption compared to graph transformers and state-of-the-art heterophily-based methods.
>
>
> *(Question 2) “Challenging heterophilic datasets”*
>
> In order to ensure a **fair and comprehensive benchmark** we considered **graphs with varying sizes** (from as low as 183 nodes up to 25’221), **densities** (from as low as 295 edges up to 2’809’204) and **homophilic ratios** (from as low as 0.09 up to 0.81). Since the Squirrel and Chameleon datasets have sizes, densities, and homophily ratios that fall well within these ranges, we chose to benchmark our model on additional datasets with more diverse characteristics to better evaluate its generalizability. However, we appreciate the reviewer’s suggestion as we will indeed, in future research, consider including such datasets to improve experimental variety.
>
>
> *(Question 3) “Minor typos”*
>
> We appreciate the reviewer’s observations and confirm that all minor issues highlighted in the review are solved in the last uploaded version of the manuscript.
>
>
> We thank the reviewer for the excellent questions and hope the revised paper addresses their concerns. We look forward to further engagement and hope this encourages an improved score.

---

> > ### Comment · Reviewer_e676 · 2024-11-28
> >
> > Thank you to the authors for providing a detailed response. However, several of my concerns remain unaddressed:
> >
> > 1. DeltaGNN is proposed as a scalable approach capable of handling large graphs. While experiments were conducted on graphs with sizes ranging from 183 nodes to ~24,000 nodes, the evidence provided is insufficient to substantiate its scalability claims. Experimental validation on significantly larger graphs is necessary to support this aspect.
> >
> > 2. As demonstrated in prior works like GPRGNN and PPGNN ([3,4] from the previous response), graph filters can be interpreted as a form of graph rewiring and have been shown to address oversmoothing effectively. Positioning the proposed method in relation to these graph filter-based methods and including direct comparisons is essential to establish its contributions.
> >
> > 3. The experiments section includes comparisons with outdated baselines. In my earlier feedback, I highlighted a more recent state-of-the-art method, ACM-GCN+/++ ([1]), which outperforms both H2GCN and DeltaGNN. Comparative evaluations with such baselines are required to provide a fair assessment of the method's performance.
> >
> > 4. While datasets like Texas, Wisconsin, and Cornell are frequently used in heterophilic GNN studies, their small graph sizes and low homophily scores limit the generalizability of the results. Moreover, simple baselines like MLP often achieve comparable or superior performance to DeltaGNN on these datasets. Additional experiments on larger, more complex heterophilic datasets such as Squirrel and Chameleon, or similar datasets, are required to strengthen the empirical evidence.
> >
> > Given these unresolved concerns, I will maintain my current rating of 5.

---

### Official Review · Reviewer_aCWG · 2024-11-04

**Soundness:** 3
**Presentation:** 2
**Contribution:** 3
**Rating:** 6
**Confidence:** 3

**Summary:**

This work targets the prevalent issues of over-smoothing and over-squashing in GNNs. It highlights that current approaches often face challenges such as high computational complexity and lack of generalizability. To tackle these issues, the authors introduce a mechanism termed 'information flow control', which employs an innovative metric known as the 'information flow score'. This mechanism is designed to mitigate over-smoothing and over-squashing while maintaining linear computational overhead. Empirical evaluations demonstrate its superior performance under constrained computational conditions.

**Strengths:**

1. The suggestion in Lemma 1—that identifying nodes connected by heterophilic edges through measuring feature differences during message aggregation— appears to be constructive.
2. The introduction effectively outlines the problems of over-smoothing and over-squashing, and provides a comprehensive overview of existing methods aimed at resolving these challenges.
3. The proposed method for addressing the problems of over-smoothing and over-squashing is both innovative and promising. The approach involves decoupling the original graph into a homophilic subgraph and a heterophilic subgraph using the proposed information flow score. Subsequently, the method performs dual aggregation on these subgraphs to capture both short-term and long-term dependencies.
4. The complexity of method information flow score is superior to those of other rewiring methods.
5. The proposed method demonstrates strong performance in terms of prediction accuracy and scalability.

**Weaknesses:**

1. "$\triangle^t_u$ can be interpreted as the velocity at which the node embeddings are aggregated at layer t." The concept of aggregation velocity is somehow confusing. More background knowledge and explanation, also examples, are required to help readers to understand the  measurement of aggregation velocity.

2. The authors propose using $(\triangle^2)^t_u = d(\triangle^t_u - \triangle^{t-1}_u)$ to measure the rate of change in the rate at which node embeddings are aggregated. However, $\triangle^t_u$ and $\triangle^{t-1}_u$ are outputs from different layers and thus belong to different spaces. Therefore, the rationale for measuring the distance between points in these two spaces is questionable. Please provide justification for this measurement.

3. The Information Flow Control (IFC) mechanism is a core component of the proposed method. Therefore, the implementation details of the IFC mechanism, including the score hill ascent framework, should be included in the main text rather than in the appendix. As currently presented, the score hill ascent framework is difficult to follow.

4. In Figure 2, some subgraphs are difficult to interpret. For example, the 'feature density - feature value' plot and the 'score - node' plot could benefit from additional clarification or improved labeling. What do the different curves in the feature density - feature value plot represent? Additionally, the phrase 'and enhance the graph score' lacks clarity. A definition of the term 'graph score' would be helpful.

5. The proof of Lemma 1 is difficult to follow. Specific issues are detailed in the following list. **Additional background information and explanation are needed to help readers understand the proof**.
   - In line 727, the term 'valid' is used to ensure that the assignment respects the given homophily ratio $\mathcal{H}_u$. However, the concept of 'valid' is not clearly defined, and it is unclear how this term ensures compliance with the specified homophily ratio. Additional background information and explanation are needed to help readers understand these aspects.
   - The relationship between $\triangle^t_u$ and the valid assignment $s$ is not explained.
   - In the equation $U(\mathcal{H}_u)_u = \operatorname{max}_{s\in S}(\triangle^t_u)$, the representation of $U$ is unclear.
   - Due to the lack of clarity, it is not possible to understand why 'any node $u$ with $\triangle^t_u > p$ will have $\mathcal{H}_u < \mathcal{H}$.'

1. The phrase 'as this quantity depends on the homophily of the node $u$', in line 727,  requires clarification. It is not immediately apparent why this quantity should depend on the homophily of the node. A clear explanation is needed to elucidate this dependency.

1. Mirror issues: a) in line 723, should "neighbourhood $N(u)$" be revised to "neighbourhood $\mathcal{N}(u)$" to consist to notation of neighborhood? b) $\bigoplus\limits_{v \in \mathcal{N}(u)}\mathbf{M}_u$ should be revised to $\bigoplus\limits_{v \in \mathcal{N}(u)}\mathbf{M}_v$.

**Questions:**

1. In Table 1, it is evident that the Information Flow Score (IFS) method underperforms other rewiring methods when combined with the GIN model, unlike with other models. This discrepancy may be due to the fact that GIN uses sum aggregation, whereas other models typically use weighted mean aggregation. The sum aggregation in GIN likely results in a higher variance for $\{\sum\limits_{v \in \mathcal{N}(u)}\mathbf{M}_v \mid u \in \mathcal{V}\}$ compared to $\{\mathbf{M}_u \mid u \in \mathcal{V}\}.$  Consequently, the so-called 'aggregation velocity' depends not only on node features but also on node degrees. This suggests that the proposed method may not be well-suited for models that use sum aggregation. Is my understanding correct?

---

> ### Author Response · Authors · 2024-11-25
>
> We sincerely thank the reviewer for their thoughtful feedback and appreciate that you found our ideas to be both novel and impactful. Below, we provide detailed responses to your comments.
>
> *(Weakness 1-2): “Node embeddings”*
>
> The concept of aggregation velocity is the physical interpretation of the rate of change of the node embeddings throughout the aggregation within a GNN layer. **The higher this number, the faster a node embedding moves within the feature space at a certain time t**. To give a graphical intuition we included a numerical experiment on a small graph of CT scans of liver tumours extracted from a MedMNIST medical dataset, where we depict how the first differences of the embeddings $\Delta^t_u$ and second differences $(\Delta^2)^t_u$ vary throughout the 4th layer DeltaGNN architecture (see Appendix D.2).
>
> $\Delta^t_u$ is assumed to be a non-negative (non necessarily monotonically decreasing) sequence converging to zero defined over $\mathbb{R}$ (this has also been clarified in Lemma 1). When we compute $(\Delta^2)^t_u$ we simply measure how this sequence varies over time, as a result, the distance between $\Delta^{t}_u$ and $\Delta^{t-1}_u$ is mathematically well defined. However, arguing whether this is conceptually correct is considerably more challenging; we appreciate this insightful objection. While in general, directly comparing node features belonging to different spaces is conceptually wrong, if we determine some **invariant** belonging to the architecture which is preserved throughout the layers, it is fair to compare how a related property changes between these. In this case, we know that the sequence $\Delta^t_u$ converges to zero for some GNN architectures and this holds for all the layer’s feature spaces. **As a result, comparing how this property changes over time is conceptually consistent**. This approach is fairly common in literature when we want to understand how a feature-dependent property changes between different layers or architectures, for instance, when measuring how feature separability changes throughout a neural network’s layers [1].
>
>
> *(Weakness 3): “Implementation of IFC”*
>
> We agree that providing implementation details in the main text would enhance clarity and accessibility for readers. We have now moved that section to the main body of the paper. With the updated version the functioning of the score hill ascent should also be more intuitive for the reader.
>
>
> *(Weakness 4): “Figure 2 clarity”*
>
> We improved the labelling of the subgraphs of Figure 2. In the subgraphs, the different curves in the feature density-feature value plots highlight how feature densities of different node classes compare before and after feature disentanglement. In the paper, the term “graph score” is used to indicate the “mean node score” of the graph. Both aspects have been clarified in the paper to improve clarity and consistency in notation.
>
>
> *(Weakness 5-6): “Lemma 1”*
>
> We appreciate the detailed feedback of Lemma 1. We removed the term “valid” introducing additional mathematical notation which, in our opinion, improved clarity. Additionally, **we expanded the final part of the proof** where we state the connection between $\Delta_u^t$, $U(\mathcal{H}_u)_u$ and the node homophilic ratio.
>
>
> *(Weakness 7): “Minor issues”*
>
> We appreciate the reviewer’s observations and confirm that all the minor issues highlighted in the review are solved in the last version of the manuscript.
>
>
> *(Question 1): “IFS on sum-aggregating GNNs”*
>
> We thank the reviewer for the insightful observation. Yes, it is correct. When we use our novel metric IFS on sum-aggregating GNNs like GIN, our assumption regarding the convergence of the sequence $\Delta^t_u$ is not respected in general (see appendix B). As a result, the aggregation velocity will fail to distinguish between different node homophilic rates. Additionally, sum-based aggregations provoke substantial changes in the feature embeddings, as a result will experience high values of $\Delta_u^t$ which will produce a high $U(\mathcal{H}_u)_u = \max_{s \in S} \Delta^t_u$ and finally high threshold $\rho \geq \max_{v \in \mathcal{V}, h \in [\mathcal{H},1]} U(h_v)_v$. However, as described in the paper, **a high threshold $\rho$ results in a bad distinction between nodes with different homophilic ratios**.
>
> We thank once again the reviewer for their excellent and detailed feedback. Your insightful questions and suggestions have helped us refine our contributions further. We hope that we have fully addressed your points and concerns. Please let us know if this remains unclear or if you consider any other discussion points to be open. We are of course happy to keep engaging with the reviewer throughout the process.
>
> *References:*
>
> [1] Schilling, A., Maier, A., Gerum, R., Metzner, C. and Krauss, P., 2021. Quantifying the separability of data classes in neural networks. Neural Networks, 139, pp.278-293.

---

> > ### Comment · Reviewer_aCWG · 2024-11-26
> >
> > Thank you very much for the authors' responses first.
> >
> > In Figure 5, the values of nodes 4, 9, and 14 are not significantly lower than those of the other nodes, making them less distinguishable than the author claims.
> >
> > The assertion "if we determine some invariant belonging to the architecture which is preserved throughout the layers, it is fair to compare how a related property changes between these" does not persuade me due to the absence of substantial scientific evidence.
> >
> > As a reference, [1] introduced the Generalized Discrimination Value (GDV) as a metric to assess the separation of different data classes within each layer of a network. In calculating GDV, each feature dimension is individually z-scored to ensure that GDV is invariant to scaling and translation. However, the measurement of delta embeddings lacks this invariance to ``scaling and translation''. Consequently, the justification for using delta embedding measurements remains insufficient.

---

> > > ### Author Response · Authors · 2024-11-28
> > >
> > > Thank you very much for your responses. Below, we further discuss the reviewer’s concerns.
> > >
> > > In Figure 5, we used a re-scaled real-world graph with node embeddings representing 64x64 medical images. This is not a simple demo-like experiment, as large images such as these introduce significant noise into the process, particularly during the first aggregation. This can also be observed in the subplots of the first and second differences, where no apparent pattern is distinguishable. However, starting from $t=1$, nodes 4, 9, and 14 can be easily distinguished by their higher values of first and second differences, which are up to 100% higher compared to the other nodes in the graph. This is also evident when analyzing the node scores, as nodes 4, 9, and 14 show up to 50% lower scores relative to the graph's mean score. These differences are significant enough to clearly identify nodes located in bottlenecked areas and heterophilic connections.
> > > In this example, we set the multipliers to $m=1$ and $l=1$. However, these can be adjusted to rescale the range of node scores. Importantly, this adjustment generally does not affect the filtering process, as the ordering of the node scores is preserved as long as $m=l$.
> > >
> > > We would like to clarify that we were not directly comparing our work to [1], as our study involves different assumptions and properties. The reference was cited to demonstrate that analyzing how a feature-dependent property changes across layers is fairly common in the literature and is not generally incorrect. Our approach, as correctly noted by the reviewer, differs substantially. Unlike [1], we do not rely on the assumption of invariance to “scaling and translation.” Instead, we are **leveraging the invariant of convergence of node embeddings over time** (which holds true under specific aggregation paradigms, such as weighted mean). In other words, we assume that the limit of the first delta embeddings exists and converges to zero.
> > > The validity of extracting information by comparing features across distinct layers depends on how this information is interpreted. Since the speed of convergence of a sequence corresponds to the differences of its consecutive terms, we calculate and interpret the second delta embeddings accordingly. If this were incorrect, we would have a sequence with an undefined speed of convergence, which contradicts our hypothesis.
> > >
> > > We hope this clarifies our previous statements. Of course, we are happy to further discuss these points if the reviewer finds it appropriate.

---

### Official Review · Reviewer_ffF7 · 2024-11-05

**Soundness:** 3
**Presentation:** 4
**Contribution:** 3
**Rating:** 5
**Confidence:** 4

**Summary:**

This paper proposes DeltaGNN which considers the long-range node interaction via information flow control or semi-supervised node classification.
The key idea is to take use of first delta embeddings $\Delta_u^t$ and the variance of second delta embeddings $\mathbb{V}_t[\Delta_u^2]$.
If the node is connected with the same labels, then $\Delta_u^t$ tends to be some, since the features from neighbors are close to center embeddings.
If the node works as a bottleneck, then the aggregated features in each layer will have a huge difference, which might cause the big variance of $(\Delta^2)_u^t$, denoted as $\mathbb{V}_t[\Delta_u^2]$.
Based on this, information flow score is used to measure the nodes that is responsible for over-smoothing and over-squashing.
Then, with graph filtering on edges, the graph cuts edges to increase the homophily for short-range interaction, and connect the selected components for long-range interactions.
Combined with these two, the author proposes the DeltaGNN.

**Strengths:**

1. This paper propose an interesting idea to connect first and second delta embedding with over-smoothing and over-squashing.
	The proposed two lemma demonstrate the relationship between them.
	Such relationship provides insight for developing algorithm to measure and alleviate over-smoothing and over-squashing problems considering the node embeddings.
2. The proposed metric inforation flow score can numerically find the nodes that might cause over-smoothing and over-squashing in this graphs.

**Weaknesses:**

1. While it is good to have a numerical metric to identity the key nodes and edges for over-smoothing and over-squashing.
	The connection of heterophilic graphs in DeltaGNN to these two metric is not that strong.
	First, it seems like heterophilic graphs is not related to solve the over-smoothing or over-squashing problem.
	Second, if using informative flow control can perfectly solve the over-smoothing or over-squashing problem,
	why model can not get perfect results?
	In other words, why heterophilic graph is needed in this case?
	Does the introduction of heterophilic graph will cause further questions about non-existing interactions?
	This part needs to be further justified. The motivation and experiments of the reasons to use this part need to be provided.

2. As a suggestion, some numerical experiments can be provided to demonstrate that with the information flow control,
	the new homophilic graph can have less over-smoothing or graph bottleneck issues on real-world datasets.
	For example, the homophilic ratio of a node can be calculated and compared between the original graphs and the rewired graphs.

3. The experiments is a little weak, and more and larger graph datasets should be included like ogb datasets.

**Questions:**

1. What is the formulation of equation of $\Theta_t\left(\mathbf{A}^{t-1}, K(t, \theta)\right.$, Score $\left.^t\right)$ that is used to filter the graph?

---

> ### Author Response · Authors · 2024-11-25
>
> We appreciate that the reviewer found our work interesting and our ideas novel. Below, we address the questions and concerns raised, which we believe have strengthened our work.
>
> *(Weakness 1) “The rationale behind heterophilic graphs.”*
>
> We thank the reviewer for the interesting questions as these are very important and interesting points you made. We would like to clarify that the purpose of the proposed DeltaGNN is to demonstrate the effectiveness of our novel approach called Information Flow Control (IFC) by introducing a scalable framework for **long-range and short-range interaction detection**. Moreover, the proposed IFC score has many applications. On the other hand, the IFC module is responsible for preventing over-smoothing and over-squashing, as demonstrated by our theoretical results (Appendix B, C) and numerical experiments (Appendix D). Moreover, as highlighted in the manuscript, the accuracy of our metric in distinguishing between homophilic and heterophilic nodes is correlated to “the maximum distance between the subspaces of M associated with the node classes” (where M is the feature space). Feature separability is domain-specific and more importantly dataset-specific, and prevents our method from getting perfect results. A more comprehensive discussion on this matter can be found in Appendix G, where we analyse the change in graph homophily ratio during the edge-filtering showing that our score can improve the graph homophilic ratio by +8.65% (Table 8). While this is far from being a perfect result, our approach proves to be **superior to all common graph topological measures** (while being less computationally expensive - Table 3) and represents an important step towards understanding how over-smoothing and over-squashing can be prevented from affecting GNNs accuracy.
>
> The presence of heterophilic graphs within DeltaGNN ensures the detection of long-range interactions and is not responsible for solving either over-smoothing or over-squashing (see Figure 3). Throughout our work, we consider exclusively node classification tasks, since our theoretical results assume the existence of node labels. In this specific task, the presence of non-existing node interactions within the heterophilic graph do not raise concerns as we are not trying to predict graph-level information (adding new edges is a fairly common technique to improve GNN node classification accuracy [1]). The learned features of the heterophilic graph detect how distant areas of the graph interact and are an essential component of our model, as it is supported by the lower classification accuracy of the ablated versions of DeltaGNN (see the performance of “GCN + rewiring(IFS)” in Table 1). We hope this is useful to clarify this point.
>
>
> *(Weakness 2)  “Numerical evidence.”*
>
> We thank the reviewer for the suggestion. Indeed, the requested experiment can be found in Appendix G, where we directly compare several topological measures (including our novel Information Flow Score - IFS) and analyse the change of the graph homophilic ratio during the edge-filtering in the CORA dataset. The **IFS shows the highest increase in the homophilic rate**, with results reported below.
>
>
> | **Methods**               | **Original Graph** | **Homophilic Filtered Graph** | **Δ Homophilic Rate** |
> |----------------------------|--------------------|-------------------------------|------------------------|
> | Degree Centrality          | 0.6676            | 0.7115                        | +6.72%                |
> | Eigenvector Centrality     | 0.6676            | 0.6937                        | +4.05%                |
> | Betweenness Centrality     | 0.6676            | 0.7057                        | +5.85%                |
> | Closeness Centrality       | 0.6676            | 0.7043                        | +5.64%                |
> | Forman-Ricci Curvature     | 0.6676            | 0.7094                        | +6.40%                |
> | Ollivier-Ricci Curvature   | 0.6676            | 0.7112                        | +6.67%                |
> | Information Flow Score     | 0.6676            | 0.7244                        | **+8.65%**            |
>
>
> Additionally, in Appendix D we included a numerical experiment on a small graph of CT scans of liver tumours extracted from a MedMNIST medical dataset, where we analyse how the connectivity of the graph and the node scores change during the edge-filtering (see Figure 5, 6). **This evidence not only supports the effectiveness of our approach in alleviating over-squashing but also its generalizability in domains which are distant from commonly tested areas, such as citation networks**, and include **large and noisy node features** [1].
>
> *References*
>
> [1] Nguyen, K., Hieu, N.M., Nguyen, V.D., Ho, N., Osher, S. and Nguyen, T.M., 2023, July. Revisiting over-smoothing and over-squashing using ollivier-ricci curvature. In International Conference on Machine Learning (pp. 25956-25979). PMLR.

---

> ### Author Response · Authors · 2024-11-25
>
> *(Weakness 3)  “New experiments on large datasets.”*
>
> We agree with the reviewer that additional experiments on larger datasets would be beneficial for further supporting our strong results on varying datasets with different disparate topologies. Unfortunately, due to our limited hardware capacity, as many ICLR research groups, we struggle to properly tune and fit GNN models in datasets with hundreds of thousands of nodes in a reasonable time frame. **For fairness, we believe that this request should not deter reviewers from acknowledging the merits of such works when they are evaluated on small-to-large scale graph data —but not very large ones such as ogb**. We have already shown that our Delta GNN produces consistently better performance on graph datasets with varying sizes, topologies, and homophily. Specifically, we showed a reproducible and consistent reduction of the average epoch time of >30% when using DeltaGNN, proving its scalability with datasets of varying sizes and edge densities. **Specifically, we benchmarked our model on 10 datasets with up to 23660 nodes and 2’809’204 edges** (which is still considerably higher than some ogb datasets, like ogb-arxiv which includes on average graphs with ~1,166,243 edges).
>
>
> *(Question 1) “Implementation of the edge filtering.”*
>
> $\Theta_t\left(\mathbf{A}^{t-1}, K(t, \theta)\right.$, Score $\left.^t\right)$ indicates an edge-filtering operation filtering K(t,\theta) edges based on the Information Flow Score at time t (Score^t). In our experiments, we offered two implementations for K(t,\theta), one **constant** (which will result in a fixed number of edges filtered at every layer of the architecture), and one **linear** (where the number of filtered edges increases with the depth of the architecture). In both cases, we removed the edges associated with the lowest scores. (additional details on the implementation can be found in Appendix F.1).
>
> We thank once again the reviewer for the excellent questions and hope that we have clarified doubts in the new version of the paper. We hope that this is able to convince the reviewer to increase their score. We are of course happy to keep engaging with the reviewer throughout the process.

---

> ### Comment · Reviewer_ffF7 · 2024-12-02
> **Reply to the Authors**
>
> Thanks for your effort for the reply, and I will keep my score.

---

### Meta-Review · Area_Chair_BBGs · 2024-12-20

**Metareview:**

**(a) Scientific Claims and Findings:**
The paper introduces a novel mechanism called information flow control (IFC) to address challenges in Graph Neural Networks (GNNs), particularly over-smoothing and over-squashing. These issues hinder the model's expressiveness and its ability to capture long-range node interactions. The authors propose a new connectivity measure, the information flow score, which operates with linear computational overhead and is supported by theoretical evidence. They present DeltaGNN, a scalable and generalizable approach for detecting both long-range and short-range interactions. Empirical evaluations across 10 real-world datasets with varying sizes, topologies, densities, and homophilic ratios demonstrate that DeltaGNN achieves superior performance with limited computational complexity.


**(b) Strengths:**
* Innovative Mechanism: The introduction of information flow control addresses critical challenges in GNNs, offering a new perspective on managing information propagation.
* Scalability and Generalizability: DeltaGNN is designed to be both scalable and generalizable, making it applicable to a wide range of graph structures and sizes.
* Empirical Validation: Evaluations across diverse (but relatively small scale) datasets provides strong empirical support for the proposed method's effectiveness. Performance improvements are in the range that can be expected by graph rewiring.
* Presentation: The paper is well written, clear, and provides good overview figures.
* Theoretical support for DeltaGNN to increase the homophily of a learning task and that the variance of Delta is related to bottlenecks.

**(c) Weaknesses:**
* Computational Complexity: While the method claims linear computational overhead, the actual computational demands, especially for large-scale graphs, are not thoroughly discussed.
* Experiments on larger scale graphs were requested but authors could not provide these experiments (even during the extended rebuttal period).
* Comparative Analysis: The paper would benefit from a more detailed comparison with existing state-of-the-art methods to highlight the advantages and potential limitations of DeltaGNN.

**(d) Reasons for Rejection:**
After careful consideration, the decision to reject the paper is based on the following reasons:
1. Insufficient Computational Analysis: The paper lacks a comprehensive analysis of the computational complexity, particularly for large-scale graphs (as highlighted by Reviewers ffF7, e676) and more challenging heterophiliy (Reviewer esWr), which is crucial for assessing the practical applicability of the proposed method.
2. Limited Theoretical Elucidation: The theoretical aspects of the information flow score could be more detailed and related to the improvements of GNN performance.
3. Need for Comparative Evaluation: A more thorough comparison with existing methods (like GPRGNN and PPGNN) is necessary to clearly establish the advantages and potential limitations of DeltaGNN.
Addressing these concerns would strengthen the paper and enhance its contribution to the field.

**Additional Comments On Reviewer Discussion:**

Most reviewers agree that more comprehensive experiments would be needed to demonstrate the merit of the proposed DeltaGNNs.

---

### Decision · Program_Chairs · 2025-01-22

Reject